# Hybrid Reinforcement Learning in Adversarial Markov Decision Processes

**Duo Cheng** [1]  **Xingyu Zhou** [2]  **Bo Ji** [1]

## Abstract

We study hybrid reinforcement learning (RL) in adversarial Markov Decision Processes (MDPs), where the learner simultaneously receives on-policy feedback from the executed policy and off-policy feedback from a fixed behavior policy, and loss functions can change arbitrarily over time. On-policy feedback allows exploration and ensures the worst-case guarantee against any comparator policy, while off-policy feedback provides coverage-dependent guarantee that scales with the "mismatch" between the behavior and comparator policies (called *coverage ratio*) and can be sharper than on-policy results whenever this ratio is small. We propose a new hybrid RL framework that accommodates adversarial losses and unknown transitions, preserving off-policy guarantees while ensuring non-trivial worst-case performance.

## 1. Introduction

Reinforcement Learning (RL) (Sutton et al., 1998) provides a general framework for sequential decision-making under uncertainty. A learner interacts with the environment over time, repeatedly observing states, selecting actions, and receiving feedback to refine future decisions, with the goal of maximizing long-term cumulative return. This formulation captures broad real-world problems, from robotics and autonomous systems (Kober et al., 2013) to online recommendation (Afsar et al., 2022), and more recently large language model alignment (Ziegler et al., 2019; Ouyang et al., 2022; Rafailov et al., 2023; Zeng et al., 2025).

A large body of the RL literature studies *on-policy* learning, where the algorithm updates its policy using feedback generated by the policy it executes in the environment (Singh et al., 2000). In each episode, the algorithm deploys a policy, observes the resulting trajectory, and updates its policy accordingly. The main challenge lies in balancing exploration and exploitation. With appropriate designs, on-policy RL admits provable performance guarantees, typically defined relative to a *comparator policy* that serves as an analytical benchmark and is often chosen as the optimal policy within a class (Shani et al., 2020; Jin et al., 2020a; Foster et al., 2021; Zhang et al., 2025).

In contrast, off-policy learning uses data generated by a behavior policy distinct from the executed policy, offering greater flexibility in feedback collection (Fujimoto et al., 2019). However, its potential performance gains over on-policy methods degrade when there is a large mismatch between behavior and comparator policies (i.e., when the "coverage ratio" is large) (Li et al., 2024).

Relying exclusively on either on-policy or off-policy feedback is fundamentally limiting: on-policy methods provide uniform guarantees against any comparator policy but can use feedback from the executed policy only, whereas off-policy methods can leverage abundant data but become inefficient when the behavior and comparator policies are highly mismatched. This trade-off motivates a *hybrid* RL paradigm that combines both, using on-policy feedback to safeguard worst-case performance while exploiting off-policy feedback to obtain sharper guarantees whenever possible. By integrating these complementary strengths, hybrid RL offers a robust and efficient learning framework.

While most existing hybrid RL work focuses on the stochastic setting, where the environment follows a stationary distribution (Xie et al., 2021; Tan et al., 2024), the more general adversarial (or non-stochastic) regime remains underexplored. In stochastic environments, one can combine on-policy and off-policy feedback by leveraging stationarity. However, many sequential decision-making problems are inherently non-stochastic (Jagerman et al., 2019), with losses varying arbitrarily (or sometimes even adversarially) over time. In such settings, pre-collected off-policy data can quickly become misaligned with the current environment, and it may no longer be a reasonable way to utilize behavior policy, while relying solely on on-policy feedback discards potentially informative auxiliary off-policy data that remain available and relevant even when losses are non-stochastic.

This tension motivates the study of hybrid RL in the adversarial regime, where the learner must be robust to arbitrary

---

[1]Department of Computer Science, Virginia Tech [2]Department of Electrical and Computer Engineering, Wayne State University. Correspondence to: Duo Cheng <duocheng@vt.edu>.

*Proceedings of the $43^{rd}$ International Conference on Machine Learning*, Seoul, South Korea. PMLR 306, 2026. Copyright 2026 by the author(s).

| Setup | On-Policy | Off-Policy | Hybrid (**Ours**) |
|---|---|---|---|
| MAB | $\sqrt{AT}$ (Audibert & Bubeck, 2009) | $\mathcal{C}(\pi^{\mathrm{C}})\sqrt{T}$ (Gabbianelli et al., 2023) | $\sqrt{T} \cdot \min\{A, \sqrt{\mathcal{C}(\pi^{\mathrm{C}})}\}$ |
| MDP (Known Transition) | $H\sqrt{SAT}$ (Zimin & Neu, 2013) | $\mathcal{C}(\pi^{\mathrm{C}})\sqrt{HT}$ (Bacchiocchi et al., 2024) | $\sqrt{T} \cdot \min\{HSA, \sqrt{H\mathcal{C}(\pi^{\mathrm{C}})}\}$ |
| MDP (Unknown Transition) | $H^2 S\sqrt{AT}$ (Jin et al., 2020a) | $H\mathcal{C}(\pi^{\mathrm{C}})\sqrt{SAT}$ (**Ours**) | $\sqrt{T} \cdot \min\left\{H^3 S^2 A, H\sqrt{SA\mathcal{C}(\pi^{\mathrm{C}})} + \mathcal{C}(\pi^{\mathrm{C}}) + H^{3/2}\sqrt{S\mathcal{C}(\pi^{\mathrm{C}})}\right\}$ |

*Table 1.* A summary of relevant results. Log. terms and constants are omitted. All bounds are universal on $\mathbb{E}[R(\pi^{\mathrm{C}})]$ in Eqs. (1) or (10).

loss sequences while leveraging off-policy feedback whenever it is useful. Addressing this key challenge requires both algorithmic and analytical tools beyond those developed for existing stochastic hybrid RL frameworks, which constitute the main contributions of this work:

- We study hybrid RL under adversarial losses by identifying a fundamental incompatibility between existing designs for stochastic environments and those for adversarial ones, due to lack of stationarity. To resolve this, we introduce a new framework that reinterprets hybrid feedback as *online learning over multiple base algorithms*, where each base algorithm corresponds to a single feedback type (i.e., either on-policy or off-policy), building on the CORRAL paradigm (Agarwal et al., 2017). Our design achieves a sharp, coverage-dependent regret whenever the coverage ratio is small, while simultaneously retaining $\sqrt{T}$-optimal worst-case guarantees under arbitrarily-large mismatch.

- We also present the first algorithm that achieves *single-policy concentrability* (i.e., guarantees whose mismatch dependence is measured only with respect to a *single behavior policy* rather than the worst case over the entire policy class) in MDPs with *unknown* transitions using off-policy feedback. This resolves a recent open question posed by Bacchiocchi et al. (2024) and provides a necessary subroutine for hybrid learning.

Table 1 summarizes relevant results and those implied by our main results. Specifically, we establish the first off-policy regret bound for MDPs with unknown transitions based on a single-policy coverage ratio; our hybrid approach achieves regret bounds that simultaneously exhibit coverage-dependent behavior and worst-case guarantees, a property previously unattained in adversarial bandit or MDP settings.

## 2. Related Work

Due to page limits, we summarize the most relevant works here and defer a comprehensive discussion to Appendix B.

**On-Policy RL** has been extensively studied in both stochastic and adversarial MDPs, where principled exploration

enables worst-case regret and/or sample complexity guarantees (Azar et al., 2017; Jin et al., 2018; Zimin & Neu, 2013; Jin et al., 2020a; Luo et al., 2021).

**Off-Policy RL** is primarily developed in stochastic settings, where performance critically depends on coverage conditions between the behavior and comparator policies (Yin et al., 2021; Jin et al., 2021b; Zhan et al., 2022; Foster et al., 2022; Li et al., 2024). In the adversarial regime, Gabbianelli et al. (2023) initiate off-policy learning in bandits, and Bacchiocchi et al. (2024) extend it to MDPs with known transitions, leaving the unknown-transition case open.

**Hybrid RL** has been studied mainly in stochastic environments, beginning with "policy finetuning" (Xie et al., 2021) and followed by several extensions (Wagenmaker & Pacchiano, 2023; Li et al., 2023; Tan et al., 2024), but remains unexplored under adversarial losses. Our work bridges these gaps by studying hybrid RL in adversarial MDPs with unknown transitions, connecting coverage-dependent guarantees with worst-case robustness.

## 3. Multi-Armed Bandits

In the remainder of the paper, we first formalize hybrid learning in the adversarial setting and introduce our core ideas in the Multi-Armed Bandit (MAB) setup (Section 3), which can be viewed as a special case of MDPs without state transitions, yet retaining core technical challenges from on/off-policy feedback. We then present our main results for the more general MDP setting (Section 4).

### 3.1. Preliminaries of Adversarial MAB

**Adversarial MAB.** An adversarial MAB is defined as a tuple $\{T, \mathcal{A}, \{\ell_t\}_{t=1}^T\}$, where $T$ is the number of rounds, $\mathcal{A}$ is the action space, and $\ell_t : \mathcal{A} \to [0, 1]$ is the loss function in round $t$. Let $A := |\mathcal{A}| < \infty$ be the finite number of actions. We assume that loss functions $\ell_t$'s are determined *before* the interaction begins and are unknown to the learner.

**Interaction Protocol.** The learner interacts with the MAB instance for $T$ rounds sequentially. In the $t$-th round, the learner determines some (randomized) policy $\pi_t : \mathcal{A} \to [0, 1]$, in which $\pi_t(a)$ denotes the probability of choosing

action $a$. Then, the learner chooses action $a_t \in \mathcal{A}$ with probability $\pi_t(a_t)$ and proceeds to the next round. Let $\Pi$ be the set that contains all feasible randomized policies.

Next, we define two feedback models in the MAB setup.

**On-Policy Feedback.** The learner observes $\{a_t, \ell_t(a_t)\}$ (i.e., feedback corresponding to the action $a_t$ from *executed* policy $\pi_t$) at the end of round $t$. This is essentially the standard "bandit feedback" in the literature.

**Off-Policy Feedback.** The learner observes $\{a_t^{\mathrm{B}}, \ell_t(a_t^{\mathrm{B}})\}$ at the end of round $t$, where $a_t^{\mathrm{B}}$ is sampled from a fixed *behavior policy* $\pi^{\mathrm{B}} \in \Pi$. Intuitively, behavior policy can be viewed as a fixed, deployed policy, which runs *in parallel* to the learner and generates one action–loss pair *online per round*. As in Gabbianelli et al. (2024); Wang et al. (2024); Ryu et al. (2025), we assume that $\pi^{\mathrm{B}}$ is known to the learner and losses $\ell_t$'s are generated *independently* of $\pi^{\mathrm{B}}$. Note that: 1) "adversarial" refers to *non-stochastic* losses rather than a malicious environment; 2) unknown $\pi^{\mathrm{B}}$ can be handled via empirical estimates (Gabbianelli et al., 2023, Sec. 3.2).

**Regret.** For any fixed policy $\pi \in \Pi$, its (expected) loss in the $t$-th round is $\langle \pi, \ell_t \rangle := \sum_{a \in \mathcal{A}} \pi(a)\ell_t(a)$. An (online learning) *algorithm* iteratively maps the history feedback to a policy. The (expected) *regret* of an algorithm against any fixed comparator policy $\pi^{\mathrm{C}} \in \Pi$ is defined as

$$\mathbb{E}[R(\pi^{\mathrm{C}})] := \mathbb{E}\left[\sum_{t=1}^{T} \langle \pi_t, \ell_t \rangle - \sum_{t=1}^{T} \langle \pi^{\mathrm{C}}, \ell_t \rangle\right], \quad (1)$$

which is the gap between the total (expected) loss of a policy sequence generated by the algorithm under consideration and that of $\pi^{\mathrm{C}}$. The expectation is taken with respect to the randomness from the algorithm (and also the behavior policy, if off-policy feedback is considered).[1] Let $\pi^* := \operatorname{argmin}_{\pi \in \Pi} \sum_{t=1}^{T} \langle \pi, \ell_t \rangle$ be the *optimal (fixed) policy* in hindsight.

**Additional Notations.** For any positive integer $n$, let $[n] := \{1, 2, \ldots, n\}$. For any finite set $\mathcal{X}$, let $\Delta(\mathcal{X})$ denote the set of all probability distributions over $\mathcal{X}$. For any convex regularizer function $\psi$, the associated Bregman divergence is $D_\psi(u, w) := \psi(u) - \psi(w) - \langle \psi(w), u - w \rangle$. We consider two specific regularizers in this paper: Shannon Entropy $\psi_t^{\mathrm{SE}}(x) := \frac{1}{\eta_t} \sum_i x(i) \log x(i)$ and logarithmic barrier (log-barrier) $\psi_t^{\mathrm{LB}}(x) := \sum_i \frac{1}{\eta_{t,i}} \log \frac{1}{x(i)}$, where $\eta_t$ and $\eta_{t,i}$ are learning rates specified later for the algorithms considered.

---

[1] Since our algorithms are randomized, it is generally not meaningful to focus on a single sample path. Instead, standard notions are expected regret (considered in this work) and high-probability regret, with the latter providing a stronger guarantee.

---

**Algorithm 1** EXP3-IX (Gabbianelli et al., 2023)

---

1: **Input:** learning rate $\eta^{\mathrm{IX}} > 0$
2: **Define:** $\eta_t \equiv \eta^{\mathrm{IX}}, \forall t \in [T]$
3: **Initialization:** $\pi_1(a) = 1/A, \forall a \in \mathcal{A}$
4: **for** $t = 1 : T$ **do**
5:     Observe off-policy feedback $\{a_t^{\mathrm{B}}, \ell_t(a_t^{\mathrm{B}})\}$ with $a_t^{\mathrm{B}} \sim \pi^{\mathrm{B}}$
6:     Construct $\widehat{\ell}_t(a) = \mathbb{1}\{a = a_t^{\mathrm{B}}\} \frac{\ell_t(a) - 1}{\pi^{\mathrm{B}}(a) + \eta^{\mathrm{IX}}/2}, \forall a \in \mathcal{A}$
7:     Calculate $\pi_{t+1} = \operatorname*{argmin}_{\pi \in \Delta(\mathcal{A})} \left( \langle \pi, \widehat{\ell}_t \rangle + D_{\psi_t^{\mathrm{SE}}}(\pi, \pi_t) \right)$
8: **end for**

---

## 3.2. MAB with Off-Policy Feedback

We first briefly review the results of Gabbianelli et al. (2023) and introduce a key notion called *coverage ratio*.

**Definition 1** (Coverage Ratio (Gabbianelli et al., 2023)). Given any fixed behavior policy $\pi^{\mathrm{B}}$, the coverage ratio with respect to (w.r.t.) any policy $\pi$ is defined as

$$\mathcal{C}_{\pi^{\mathrm{B}}}(\pi) := \sum_{a \in \mathcal{A}} \frac{\pi(a)}{\pi^{\mathrm{B}}(a)}. \quad (2)$$

Since $\pi^{\mathrm{B}}$ is fixed, we drop it in $\mathcal{C}_{\pi^{\mathrm{B}}}(\pi)$ and use $\mathcal{C}(\pi)$ for notational ease. The coverage ratio measures the mismatch between the comparator and behavior policies: it is small when they are close and grows as they diverge. It can be as small as $O(1)$ (e.g., when $\pi^{\mathrm{C}}(a') = 1$ and $\pi^{\mathrm{B}}(a') = 1/2$ for some $a'$) or arbitrarily large as $\pi^{\mathrm{B}}(a') \to 0$.

The main result of Gabbianelli et al. (2023) is Algorithm 1 with a $\mathcal{C}(\pi^{\mathrm{C}})$-dependent regret bound shown below.

**Theorem 1** (Theorem 1 of Gabbianelli et al. (2023)). *With only off-policy feedback, Algorithm 1 ensures that*

$$\mathbb{E}[R(\pi^{\mathrm{C}})] = \widetilde{O}\left( \frac{1}{\eta^{\mathrm{IX}}} + \eta^{\mathrm{IX}} T \mathcal{C}(\pi^{\mathrm{C}}) \right), \forall \pi^{\mathrm{C}} \in \Pi. \quad (3)$$

*Remark* 1 (Learning Rate Tuning). Based on Eq. (3), there are two ways to choose $\eta^{\mathrm{IX}}$ as in Gabbianelli et al. (2023), either $\pi^{\mathrm{C}}$-dependent or $\pi^{\mathrm{C}}$-independent. These choices incur a trade-off between bound sharpness and universality: 1) with $\eta^{\mathrm{IX}} = \frac{1}{\sqrt{\mathcal{C}(\pi^{\mathrm{C}})T}}$, we get $\widetilde{O}\left(\sqrt{\mathcal{C}(\pi^{\mathrm{C}})T}\right)$ regret but only against $\pi^{\mathrm{C}}$ (and others with the same coverage ratio); 2) with $\eta^{\mathrm{IX}} = \frac{1}{\sqrt{T}}$, we get *universal* $\widetilde{O}\left(\mathcal{C}(\pi^{\mathrm{C}})\sqrt{T}\right)$ regret, which holds simultaneously for every comparator policy $\pi^{\mathrm{C}}$ without prior knowledge of it, while the $\mathcal{C}(\pi^{\mathrm{C}})$ factor is looser than the $\sqrt{\mathcal{C}(\pi^{\mathrm{C}})}$ in the first scenario.

**Single-Policy Concentrability.** The significance of the above result is that the regret against any $\pi^{\mathrm{C}}$ depends on the coverage ratio w.r.t. $\pi^{\mathrm{C}}$ itself only, rather than the supremum over all policies. This desired phenomenon is called *single-policy concentrability* (Zhan et al., 2022), for which we give more details below.

**Biased Loss Estimator for Single-Policy Concentrability.** The key towards this property lies in Line 6 of Algorithm 1: the $\eta/2$ additive bias in the denominator of $\widehat{\ell}_t(a)$. Without this bias, $\widehat{\ell}_t(a)$ would have become the standard importance-weighted estimator in adversarial MABs, leading to a different regret bound, which depends on the *worst-case coverage*:

$$C_{\text{all}} := \sup_{\pi \in \Pi} \mathcal{C}(\pi) = \frac{1}{\min_{a \in \mathcal{A}} \pi^{\text{B}}(a)}. \tag{4}$$

This dependence arises because the analysis involves $\sum_{a \in \mathcal{A}} \frac{\pi_t(a)}{\pi^{\text{B}}(a)}$ rather than the desired $\sum_{a \in \mathcal{A}} \frac{\pi^{\text{C}}(a)}{\pi^{\text{B}}(a)}$, forcing a worst-case bound over on-the-fly policies $\pi_t$'s. Consequently, even when competing against a *well-covered* $\pi^{\text{C}}$, the regret still depends on the worst-case coverage induced by $\pi^{\text{B}}$, rather than the coverage between $\pi^{\text{C}}$ and $\pi^{\text{B}}$. This less desired phenomenon is known as *all-policy concentrability*, where low regret against a single $\pi^{\text{C}}$ requires *all* policies to be well covered, which may not easily hold in practice.

*Remark* 2. Viewing $(1 - \ell_t(a))$ as rewards, maximizing $(1 - \ell_t(a))$ is equivalent to minimizing loss $-(1 - \ell_t(a))$. Then, Algorithm 1 essentially resembles the reward-based view in Gabbianelli et al. (2023). To respect the convention in adversarial online learning, we still consider non-negative *losses* with an (implicit) conversion to rewards in our loss estimators. Such separations (between losses and rewards) are common in online learning (Cesa-Bianchi & Lugosi, 2006; Gabbianelli et al., 2024; Zhang et al., 2025).

**Benefit of Small Coverage Ratios.** Quantitatively, when $\pi^{\text{C}}$ is well covered by $\pi^{\text{B}}$ (i.e., $\mathcal{C}(\pi^{\text{C}})$ is small), the regret against $\pi^{\text{C}}$ can even beat the $\Theta(\sqrt{AT})$ on-policy minimax rate. For instance, if $\pi^{\text{C}}$ is the optimal policy with $\pi^*(a^*) = 1$, where $a^* := \operatorname{argmin}_{a \in \mathcal{A}} \sum_{t=1}^{T} \ell_t(a)$, then the regret against $\pi^*$ is $\widetilde{O}(\sqrt{T})$ (independent of $\text{poly}(A)$), whenever $\mathcal{C}(\pi^*) = \pi^*(a^*)/\pi^{\text{B}}(a^*) = O(1)$.

Despite the sharp bounds from small coverage ratios, the regret degrades and becomes linear ($O(T)$) as $\mathcal{C}(\pi^{\text{C}})$ grows (e.g., when $\pi^{\text{C}}(a') = 1$ and $\pi^{\text{B}}(a') = \Theta(1/T)$ for some $a'$). In contrast, using *on-policy feedback*, one can always guarantee $\widetilde{O}(\sqrt{AT})$ regret even against the strongest comparator policy (e.g., EXP3 (Auer et al., 2002)). These observations naturally motivate the following key question:

> With both on-policy and off-policy feedback, can one still enjoy $\mathcal{C}(\pi^{\text{C}})$-dependent regret while preserving worst-case guarantees when $\mathcal{C}(\pi^{\text{C}})$ is arbitrarily large?

### 3.3. MAB with Hybrid Feedback

Motivated by the discussion above, we study hybrid feedback in the MAB setting, where the learner observes both on-policy and off-policy feedback $\{a_t, \ell_t(a_t), a_t^{\text{B}}, \ell_t(a_t^{\text{B}})\}$.

#### 3.3.1. CHALLENGE AND INSIGHTS

While hybrid feedback has been extensively studied under stochastic losses (Xie et al., 2021; Li et al., 2023; Tan et al., 2024; Nguyen-Tang & Arora, 2024; Huang et al., 2025), those existing designs do not extend to adversarial settings where the losses are *no longer stationary*.

First, in the stochastic case, the use of behavior policy is to pre-collect a dataset offline *before* online interactions (Xie et al., 2021). Intuitively, this is because the loss distribution never changes. However, in the adversarial setting, pre-collected samples can highly deviate from the loss functions in the online stage, so the behavior policy must be used also *online*. Second, existing methods crucially exploit the stationarity of the loss distribution: confidence-bound (Wagenmaker & Pacchiano, 2023; Tan et al., 2024; Huang et al., 2025) and stage-based (Li et al., 2023; Nguyen-Tang & Arora, 2024) designs break down when losses vary over time and information cannot spread across stages.

A viable solution must therefore operate under adversarial losses while dynamically integrating on-policy and off-policy feedback. Under adversarial losses, however, the relative reliability of these feedback sources can vary arbitrarily over time, making any fixed choice of feedback sub-optimal *a priori*. This requires an aggregation mechanism that adaptively compares and combines multiple feedback/algorithms online, without loss stationarity to explicitly exploit.

#### 3.3.2. OUR DESIGN AND MAIN RESULTS

The motivations and insights above align well with the COR-RAL framework (Agarwal et al., 2017; Luo et al., 2022), which is a meta algorithm that combines multiple base algorithms and adapts to the best one online. In our setting, it combines two base algorithms: one on-policy algorithm (with index 0) that uses on-policy feedback and provides worst-case guarantees independent of $\mathcal{C}(\pi^{\text{C}})$; one off-policy algorithm (with index 1) that leverages off-policy feedback to obtain $\mathcal{C}(\pi^{\text{C}})$-dependent regret.

At the base level, each algorithm $i \in \{0, 1\}$ maintains its own policy $\pi_t^i \in \Pi$. At the meta level, CORRAL treats each base algorithm as an arm in MAB, maintaining a distribution $p_t$ over $\{0, 1\}$. In round $t$, CORRAL samples $i_t \sim p_t$, executes $a_t \sim \pi_t^{i_t}$, and then uses hybrid feedback $\{a_t, \ell_t(a_t), a_t^{\text{B}}, \ell_t(a_t^{\text{B}})\}$ to update both $p_{t+1}$ and $\pi_{t+1}^i$'s. Note that the on-policy feedback comes from the executed policy $\pi_t^{i_t}$, which is determined at the meta level.

In the analysis, we decompose the actual regret into two levels (meta and base levels) as

$$\mathbb{E}[R(\pi^{\text{C}})] = \mathbb{E}\left[\sum_{t=1}^{T} \langle \pi_t^{i_t} - \pi_t^i, \ell_t \rangle\right] + \mathbb{E}\left[\sum_{t=1}^{T} \langle \pi_t^i - \pi^{\text{C}}, \ell_t \rangle\right],$$

$$\tag{5}$$

---

**Algorithm 2** EXP3

1: **Define:** $\eta_t = \sqrt{\frac{\log(A)}{A \sum_{t'=1}^{t} \frac{1}{p_{t',0}}}}$; $\mathrm{upd}_t = 1\{i_t = 0\}$

2: **Initialization:** policy $\pi_1^0(a) = 1/A, \forall a \in \mathcal{A}$

3: **for** $t = 1 : T$ **do**

4:     Receive $\{i_t, p_{t,0}, a_t, \ell_t(a_t)\}$ from meta algorithm

5:     Construct $\widehat{\ell}_t(a) = \frac{\mathrm{upd}_t}{p_{t,0}} \cdot 1\{a = a_t\} \frac{\ell_t(a)}{\pi_t^0(a)}, \forall a \in \mathcal{A}$

6:     Calculate $\pi_{t+1}^0 = \underset{\pi \in \Delta(\mathcal{A})}{\mathrm{argmin}} \left(\langle \pi, \widehat{\ell}_t \rangle + D_{\psi_t^{\mathrm{SE}}}(\pi, \pi_t^0)\right)$

7:     Suggest policy $\pi_{t+1}^0$ to meta algorithm

8: **end for**

---

and bound the regret in each level separately. With an appropriate update rule for $p_t$'s, CORRAL guarantees that the actual regret $\mathbb{E}[R(\pi^{\mathrm{C}})]$ is close to that of *any* base algorithm *run in isolation*. Next, we present our full hybrid learning algorithm and establish its theoretical guarantees.

**On-Policy Base Algorithm.** Our on-policy base algorithm is (a variant of) the well-known EXP3 that ensures $O(\sqrt{AT \log A})$ regret against any $\pi^{\mathrm{C}}$ with on-policy feedback alone. The variant we use is from Dann et al. (2023), whose pseudo-code is given in Algorithm 2. Compared to its vanilla version (Auer et al., 2002), the only modification under CORRAL is in the loss estimator. Let $\mathrm{upd}_t$ indicate whether the on-policy base algorithm is sampled by CORRAL at round $t$ (which occurs with probability $p_{t,0}$). If $\mathrm{upd}_t = 1$, the executed action $a_t \sim \pi_t^0$ gets loss estimate $\frac{\ell_t(a_t)}{p_{t,0} \cdot \pi_t^0(a_t)}$, while all others receive zero; if $\mathrm{upd}_t = 0$, all loss estimates are zero, and $\pi_{t+1}^0 = \pi_t^0$.

Consequently, Algorithm 2 gets updated only when sampled by the meta algorithm. Its base-level regret therefore depends on the probabilities $\{p_{t,0}\}_{t=1}^T$: when these probabilities are small, updates are infrequent, and the regret against $\pi^{\mathrm{C}}$ increases, as formalized below in Theorem 2.

*Remark* 3. Algorithm 2 is (an) *on-policy* (update procedure) since its maintained policy $\pi_t^0$ is updated using feedback *associated only with itself* (when it is sampled by the meta algorithm). Generally, on-policy feedback (associated with executed policy $\pi_t = \pi_t^{i_t}$) is generated in the meta algorithm, rather than in Algorithm 2.

**Theorem 2** (Agarwal et al. (2017); Dann et al. (2023))**.** *For any $\pi^{\mathrm{C}} \in \Pi$, Algorithm 2 ensures that*

$$\mathbb{E}\left[\sum_{t=1}^T \langle \pi_t^0 - \pi^{\mathrm{C}}, \ell_t \rangle\right] = \widetilde{O}\left(\mathbb{E}\left[\sqrt{\frac{AT}{\min_{t \in [T]} p_{t,0}}}\right]\right). \quad (6)$$

**Off-Policy Base Algorithm.** For the off-policy component, we instantiate $\Theta(\log T)$ copies of Algorithm 1, each with a distinct learning rate, instead of using a single instance as in the main idea. This multi-instance design preserves the desired $\sqrt{\mathcal{C}(\pi^{\mathrm{C}})}$ dependence in the regret bound while

remaining universal in the hybrid setting.

Specifically, we construct $(\lfloor \log T \rfloor + 1)$ instances of Alg. 1 indexed by $i \in [\lfloor \log T \rfloor + 1]$, where the $i$-th one uses (base-level) learning rate $\eta^i := (\exp(i) \cdot T)^{-1/2}$ (i.e., $\eta^{\mathrm{IX}}$ in Alg. 1) and intuitively is tuned for $\pi^{\mathrm{C}}$'s whose coverage ratio is "around" $\exp(i)$. Collectively, all instances cover coverage ratios from 1 to $T$ and ensure that for each policy $\pi^{\mathrm{C}}$, at least one instance provides base-level regret of $\widetilde{O}(\sqrt{\mathcal{C}(\pi^{\mathrm{C}})T})$.[2] Note that: 1) all off-policy instances use *the same* pair $\{a_t^{\mathrm{B}}, \ell_t(a_t^{\mathrm{B}})\}$ for updates; 2) this idea originates from Zhang et al. (2018) and is widely adopted in online optimization (Zhao et al., 2021).

**Theorem 3.** *Let $\{\pi_t^i\}_{t=1}^T$ be the policy sequence of the $i$-th base algorithm. For each policy $\pi^{\mathrm{C}}$, there exists some off-policy base algorithm index $i \in [\lfloor \log T \rfloor + 1]$ such that*

$$\mathbb{E}\left[\sum_{t=1}^T \langle \pi_t^i - \pi^{\mathrm{C}}, \ell_t \rangle\right] = \widetilde{O}\left(\sqrt{\mathcal{C}(\pi^{\mathrm{C}})T}\right). \quad (7)$$

*Remark* 4. The base-level regrets of off-policy algorithms do not depend on $\{p_{t,i}\}_{t=1}^T$ under CORRAL, because off-policy feedback is available in every round, and they need not get sampled by the meta algorithm to update themselves.

**Meta Algorithm (CORRAL).** Although any off-the-shelf MAB algorithm (e.g., EXP3) can guarantee a $\widetilde{O}(\sqrt{T})$ meta-level regret against any fixed base algorithm, this does not translate into a meaningful guarantee when corralling the on-policy base algorithm. Roughly, this is due to the $\frac{1}{\sqrt{\min_{t \in [T]} p_{t,0}}}$ factor in its base-level regret. To address this, Agarwal et al. (2017) propose a novel increasing learning-rate schedule together with the log-barrier regularizer.

The full pseudo-code of the meta algorithm is given in Algorithm 3. We state its regret guarantee in Theorem 4.

*Remark* 5 (The Use of Hybrid Feedback). In our hybrid design, off-policy feedback $\{a_t^{\mathrm{B}}, \ell_t(a_t^{\mathrm{B}})\}$ is used *solely* to get $\{\pi_{t+1}^i\}_{i \geq 1}$ via off-policy base algs. (instances of Alg. 1). On-policy feedback $\{a_t, \ell_t(a_t)\}$ is used to get both $p_{t+1}$ (via CORRAL) and $\pi_{t+1}^0$ via on-policy base alg. (Alg. 2).

**Theorem 4.** *With hybrid feedback, choosing $\eta^{\mathrm{meta}} = \frac{1}{\sqrt{T}}$ in Algorithm 3 ensures the following regret bound:*

$$\mathbb{E}[R(\pi^{\mathrm{C}})] = \widetilde{O}\left(\sqrt{T} \cdot \min\left\{A, \sqrt{\mathcal{C}(\pi^{\mathrm{C}})}\right\}\right), \forall \pi^{\mathrm{C}} \in \Pi. \quad (8)$$

We conclude with several remarks: 1) against any $\pi^{\mathrm{C}}$ (even with arbitrarily large $\mathcal{C}(\pi^{\mathrm{C}})$), the regret is always $\widetilde{O}(A\sqrt{T})$. Compared with the $\widetilde{O}(\sqrt{AT})$ guarantee with on-policy feedback alone, there is an extra $\sqrt{A}$ factor, which can be viewed

---

[2]Although $\mathcal{C}(\pi^{\mathrm{C}})$ can be arbitrarily large (e.g., $\Omega(T)$ or even infinite), such cases need not be covered since the corresponding $\widetilde{O}(\sqrt{\mathcal{C}(\pi^{\mathrm{C}})T})$ regret degenerates to the trivial $\Omega(T)$.

---

**Algorithm 3** Hybrid Learning in Adversarial MAB

1: **Input:** (meta-level) initial learning rate $\eta^{\mathrm{meta}} > 0$
2: **Define:** truncated probability simplex $\Omega'(\mathcal{X}) = \{x \in \Omega(\mathcal{X}) : x(i) \geqslant 1/T, \forall i\}$; factor $\kappa = \exp(1/\log T)$; number of base algorithms $N_{\mathrm{base}} = \lfloor \log T \rfloor + 2$
3: **Initialization:** $\eta_{1,i} = \eta^{\mathrm{meta}}, \rho_{1,i} = 2N_{\mathrm{base}}, p_{1,i} = 1/N_{\mathrm{base}}, \forall i = 0, \ldots, N_{\mathrm{base}} - 1$; create $(N_{\mathrm{base}} - 1)$ instances of Alg. 1, the (base-level) learning rate of the $i$-th one (i.e., $\eta^{\mathrm{IX}}$ in Alg. 1) is $\eta^i := \frac{1}{\sqrt{\exp(i) \cdot T}}$
4: **for** $t = 1 : T$ **do**
5:     /* execute policy and get on-policy feedback */
6:     Sample base algorithm (index) $i_t \sim p_t$
7:     Execute $\pi_t = \pi_t^{i_t}$, take $a_t \sim \pi_t$, and observe $\ell_t(a_t)$
8:     /* perform base-level updates */
9:     Pass $\{i_t, p_{t,0}, a_t, \ell_t(a_t)\}$ to Alg. 2 and get $\pi_{t+1}^0$ from it
10:     **for** off-policy base alg. $i = 1, \ldots, N_{\mathrm{base}} - 1$ **do**
11:         Get $\pi_{t+1}^i$ from the $i$-th instance of Alg. 1
12:     **end for**
13:     /* perform meta-level updates */
14:     Construct $\widehat{c}_t(i) = 1\{i = i_t\}\frac{\ell_t(a_t)}{p_{t,i}}, \forall i \geqslant 0$
15:     Update $p_{t+1} = \underset{p \in \Omega'(\{0, \ldots, N_{\mathrm{base}} - 1\})}{\mathrm{argmin}} (\langle p, \widehat{c}_t \rangle + D_{\psi_t^{\mathrm{LB}}}(p, p_t))$
16:     **for** $i = 0, \ldots, N_{\mathrm{base}} - 1$ **do**
17:         **if** $\frac{1}{p_{t+1,i}} > \rho_{t,i}$, **then** $\rho_{t+1,i} = \frac{2}{p_{t+1,i}}, \eta_{t+1,i} = \eta_{t,i}\kappa$
18:         **else** $\rho_{t+1,i} = \rho_{t,i}, \eta_{t+1,i} = \eta_{t,i}$
19:         /* $\eta_{t,i}$'s are used for $p_{t+1}$ (via $\psi_t^{\mathrm{LB}}$) */
20:     **end for**
21: **end for**

---

as the cost of corralling; 2) with off-policy feedback alone (Thm. 1), the $\sqrt{\mathcal{C}(\pi^{\mathrm{C}})}$ dependence is obtained with a $\pi^{\mathrm{C}}$-dependent tuning and is *not* universal. However, with hybrid feedback, we preserve the sharp $\sqrt{\mathcal{C}(\pi^{\mathrm{C}})}$ dependence while being universal. This is achieved thanks to the on-policy feedback with which we create and corral $\Theta(\log T)$ off-policy base-level instances (via updating $p_t$); 3) with off-policy feedback alone, whether $\sqrt{\mathcal{C}(\pi^{\mathrm{C}})T}$ *universal* bound can be achieved remains open (Gabbianelli et al., 2023).

## 4. Markov Decision Process

### 4.1. Preliminaries of Adversarial MDP

**Adversarial Markov Decision Process.** An (episodic) adversarial Markov Decision Process (AMDP) is defined as a tuple $\{T, H, \mathcal{S}, \mathcal{A}, P, \{\ell_{t,h}\}_{t \in [T], h \in [H]}\}$, where $T$ is the number of episodes, $H$ is the horizon length, $\mathcal{S}$ is the state space, $\mathcal{A}$ is the action space, $P : [H] \times \mathcal{S} \times \mathcal{A} \times \mathcal{S} \to [0, 1]$ is the transition function, and $\ell_{t,h} : \mathcal{S} \times \mathcal{A} \to [0, 1]$ is the loss function with $\ell_{t,h}(s, a)$ denoting the loss of visiting state $s$ and taking action $a$ in the $h$-th step of the $t$-th episode. Let $S := |\mathcal{S}| < \infty$ and $A < \infty$ be the finite number of states and actions, respectively. Loss functions $\ell_{t,h}$'s are unknown

(to the learner) and determined before interactions. An MDP reduces to an MAB when $H = 1$ and $\mathcal{S}$ is a singleton.

**Interaction Protocol.** The learner interacts with the AMDP for $T$ episodes sequentially. In the $t$-th episode, the learner first determines some randomized policy $\pi_t : [H] \times \mathcal{S} \times \mathcal{A} \to [0, 1]$ where $\pi_{t,h}(a \mid s)$ is the probability of taking action $a$ at step $h$ given state $s$ (so that $\sum_{a \in \mathcal{A}} \pi_{t,h}(a \mid s) = 1$). Then, the learner starts from some fixed initial state $s_{\mathrm{init}} \in \mathcal{S}$ and performs $H$-step interactions: in the $h$-th step (where $h \leqslant H - 1$), the learner observes state $s_{t,h} \in \mathcal{S}$, chooses action $a_{t,h} \in \mathcal{A}$ with probability $\pi_{t,h}(a_{t,h} \mid s_{t,h})$, and then transits to state $s_{t,h+1} \in \mathcal{S}$ with probability $P^h(s_{t,h+1} \mid s_{t,h}, a_{t,h})$; in the $H$-th step, the learner chooses action $a_{t,H}$ and deterministically transits to some (fixed) terminal state $s_{\mathrm{ter}} \notin \mathcal{S}$, which implies the end of the current episode.

Now, let $\Pi$ denote the set that contains all feasible randomized policies for MDPs. For any fixed policy $\pi \in \Pi$, we denote its expected total loss in the $t$-th episode as

$$V^{\pi,P}(s_{\mathrm{init}}; \ell_t) := \mathbb{E}\left[\sum_{h=1}^{H} \ell_{t,h}(s_{t,h}, a_{t,h}) \mid P, \pi\right]. \quad (9)$$

**On-Policy Feedback.** The learner observes $\{s_{t,h}, a_{t,h}, \ell_{t,h}(s_{t,h}, a_{t,h})\}_{h=1}^{H}$ when the $t$-th episode ends. This corresponds to the standard "bandit feedback" in MDPs. Let $1_{t,h}^{\mathrm{on}}(s, a) := 1\{s_{t,h} = s, a_{t,h} = a\}$ and $1_{t,h}^{\mathrm{on}}(s, a, s') := 1\{s_{t,h} = s, a_{t,h} = a, s_{t,h+1} = s'\}$.

**Off-Policy Feedback.** The learner observes $\{s_{t,h}^{\mathrm{B}}, a_{t,h}^{\mathrm{B}}, \ell_{t,h}(s_{t,h}^{\mathrm{B}}, a_{t,h}^{\mathrm{B}})\}_{h=1}^{H}$ when the $t$-th episode ends (where $\{s_{t,h}^{\mathrm{B}}, a_{t,h}^{\mathrm{B}}\}_{h=1}^{H}$ is the state-action pair sequence generated by behavior policy $\pi^{\mathrm{B}} \in \Pi$ under the same interaction protocol as $\pi_t$). Let $1_{t,h}^{\mathrm{off}}(s, a) := 1\{s_{t,h}^{\mathrm{B}} = s, a_{t,h}^{\mathrm{B}} = a\}$ and $1_{t,h}^{\mathrm{off}}(s, a, s') := 1\{s_{t,h}^{\mathrm{B}} = s, a_{t,h}^{\mathrm{B}} = a, s_{t,h+1}^{\mathrm{B}} = s'\}$.

**Regret.** The *regret* against any policy $\pi^{\mathrm{C}} \in \Pi$ is

$$\mathbb{E}[R(\pi^{\mathrm{C}})] := \mathbb{E}\left[\sum_{t=1}^{T}\left(V^{\pi_t,P}(s_{\mathrm{init}}; \ell_t) - V^{\pi^{\mathrm{C}},P}(s_{\mathrm{init}}; \ell_t)\right)\right]. \quad (10)$$

The expectation is taken over the randomness from the algorithm, behavior policy, and state transitions, which all together induce the random policy sequence $\{\pi_t\}_{t=1}^{T}$. Analogously, we define the optimal (fixed) policy in MDP as $\pi^* := \mathrm{argmin}_{\pi \in \Pi} \sum_{t=1}^{T} V^{\pi,P}(s_{\mathrm{init}}; \ell_t)$.

**Occupancy Measure.** We additionally introduce the notion of *occupancy measure* (Zimin & Neu, 2013). Given any policy $\pi \in \Pi$ and transition function $P'$ of an MDP, we let $q^{\pi,P'} : [H] \times \mathcal{S} \times \mathcal{A} \to [0, 1]$ denote the occupancy measure *induced* by $\pi$ and $P'$, with $q_h^{\pi,P'}(s, a)$ denoting the probability of visiting state $s$ and taking action $a$ in the $h$-th step when executing policy $\pi$ under transition $P'$.

For any transition function $P'$, we let $\mathrm{OM}(P') := \{q^{\pi,P'} : \pi \in \Pi\}$. Similarly, for any set $\mathcal{P}'$ of transition functions, we let $\mathrm{OM}(\mathcal{P}') := \{q^{\pi,P'} : P' \in \mathcal{P}', \pi \in \Pi\}$. Given any valid $q^{\pi,P'}$, one can induce policy $\pi$ via normalization

$$\pi_h(a \mid s) = \frac{q_h^{\pi,P'}(s,a)}{\sum_{s \in \mathcal{S}} q_h^{\pi,P'}(s,a)}, \forall (h,s,a) \in [H] \times \mathcal{S} \times \mathcal{A}. \tag{11}$$

With occupancy $q^{\pi,P'}$, one can rewrite $V^{\pi,P'}(s_{\mathrm{init}}; \ell_t)$ as

$$\langle q^{\pi,P'}, \ell_t \rangle := \sum_{(h,s,a) \in [H] \times \mathcal{S} \times \mathcal{A}} q_h^{\pi,P'}(s,a) \cdot \ell_{t,h}(s,a), \tag{12}$$

From the view of occupancy, loss functions become linear, which allows leveraging online convex optimization theories. For notational ease, we let $q^{\mathrm{C}} := q^{\pi^{\mathrm{C}},P}$, $q^{\mathrm{B}} := q^{\pi^{\mathrm{B}},P}$, and $q_t := q^{\pi_t,P}$. Consequently, regret can be written as

$$\mathbb{E}[R(\pi^{\mathrm{C}})] := \mathbb{E}\left[\sum_{t=1}^{T} \langle q_t, \ell_t \rangle - \sum_{t=1}^{T} \langle q^{\mathrm{C}}, \ell_t \rangle\right]. \tag{13}$$

**Coverage Ratio.** Accordingly, coverage ratios in adversarial MDPs (Bacchiocchi et al., 2024) are defined as

$$\mathcal{C}(\pi) := \sum_{(h,s,a) \in [H] \times \mathcal{S} \times \mathcal{A}} \frac{q_h^{\pi,P}(s,a)}{q_h^{\mathrm{B}}(s,a)} \text{ and } C_{\mathrm{all}} := \sup_{\pi \in \Pi} \mathcal{C}(\pi). \tag{14}$$

### 4.2. Known Transition

When true transition $P$ is known, the hybrid feedback setting admits a *straightforward extension* of MAB case by switching to the occupancy measure space $\mathrm{OM}(P)$. In this regime, we obtain a universal regret bound of order $\mathbb{E}[R(\pi^{\mathrm{C}})] = \widetilde{O}\left(\sqrt{T} \cdot \min\{HSA, \sqrt{H\mathcal{C}(\pi^{\mathrm{C}})}\}\right)$. Formal statements and proofs are deferred to Appendix D.

### 4.3. Unknown Transition

We now present our main results on hybrid feedback when the MDP transition is unknown. A prerequisite for this setting is an off-policy base algorithm that achieves $\mathcal{C}(\pi^{\mathrm{C}})$-dependent regret, as in the MAB setup. However, existing results only cover the known-transition case (Bacchiocchi et al., 2024), and whether single-policy concentrability can be achieved under unknown transitions remains open.

As a first step, we present an algorithm that attains single-policy concentrability with unknown transitions, thereby providing an affirmative answer to this open problem. We start with why existing approaches (for MABs and known-transition MDPs) do not extend to the unknown transition, which motivates new design components in this work.

#### 4.3.1. AN ATTEMPT TO ADDRESS UNKNOWN TRANSITIONS USING CONFIDENCE SETS

A standard approach for handling unknown transitions is to maintain a *confidence set* of transition functions, and then to perform online updates over the occupancies induced by *all* transitions within this set (Jin et al., 2020a). Letting $\mathcal{P}_t$ be some confidence set of transition functions maintained in episode $t$ (which contains true transition $P$), we first update the occupancy via Online Mirror Descent (OMD)

$$\widehat{q}_{t+1} = \underset{q \in \mathrm{OM}(\mathcal{P}_t)}{\mathrm{argmin}} \left(\langle q, \widehat{\ell}_t \rangle + D_{\psi_t^{\mathrm{SE}}}(q, \widehat{q}_t)\right), \tag{15}$$

and then get the corresponding policy $\pi_{t+1}$ induced by $\widehat{q}_{t+1}$. Since the update is over a transition set, regret $\mathbb{E}\left[\sum_{t=1}^{T} \langle q^{\pi_t,P} - q^{\pi^{\mathrm{C}},P}, \ell_t \rangle\right]$ is typically decomposed into

$$\underbrace{\mathbb{E}\left[\sum_{t=1}^{T} \langle \widehat{q}_t - q^{\pi^{\mathrm{C}},P}, \ell_t \rangle\right]}_{\text{OMDREG}} + \underbrace{\mathbb{E}\left[\sum_{t=1}^{T} \langle q^{\pi_t,P} - \widehat{q}_t, \ell_t \rangle\right]}_{\text{ERROR}}, \tag{16}$$

where OMDREG is the gap (in terms of total expected loss) between the output sequence $\{\widehat{q}_t\}_{t=1}^{T}$ and comparator $q^{\pi^{\mathrm{C}},P}$; ERROR is the gap between $\widehat{q}_t$ and its counterpart with true transition $P$, which is due to transition estimation error: if $P$ is known, then ERROR $= 0$, which recovers the results in Bacchiocchi et al. (2024). The first term OMDREG admits a standard OMD analysis as in previous cases: together with the biased loss estimator, one can show that OMDREG $= \widetilde{O}\left(H/\eta + \eta T\mathcal{C}(\pi^{\mathrm{C}})\right)$. However, since ERROR term depends on $\{\pi_t\}_{t=1}^{T}$ rather than $\pi^{\mathrm{C}}$, by the standard analysis (Jin et al., 2020a), one can only relate it to $\sum_{t=1}^{T} \mathcal{C}(\pi_t)$, which leads to a $\mathcal{C}_{\mathrm{all}}$ dependence.

**Lack of Pessimism.** Running OMD over transition sets fails because this approach inherently implements *optimism* (Jin et al., 2020a), which *conflicts* with the goal of introducing *pessimism* in off-policy learning. Such updates *cannot be directly modified to enforce pessimism* (e.g., unlike simple sign flips from upper to lower confidence bound (Jin et al., 2021b)), thereby motivating a new algorithmic design.

#### 4.3.2. ACHIEVING SINGLE-POLICY CONCENTRABILITY VIA BIASED TRANSITION FUNCTION

In this section, we show that, by further incorporating the *biased transition function* in Jin et al. (2023) (originally developed for an entirely different purpose), we can achieve $\mathcal{C}(\pi^{\mathrm{C}})$-type bound even when the transition is unknown. Specifically, instead of maintaining a confidence set $\mathcal{P}_t$, we construct a single transition $\widetilde{P}_t$ (see later for its properties and exact calculations), and then perform OMD update as

$$\widehat{q}_{t+1} = \underset{q \in \mathrm{OM}(\widetilde{P}_t)}{\mathrm{argmin}} \left(\langle q, \widehat{\ell}_t \rangle + D_{\psi_t^{\mathrm{SE}}}(q, \widehat{q}_t)\right). \tag{17}$$

For this update rule, we decompose the regret into

$$\mathbb{E}\left[\sum_{t=1}^{T}\langle q^{\pi_t, P} - q^{\pi^C, P}, \ell_t\rangle\right] = \underbrace{\mathbb{E}\left[\sum_{t=1}^{T}\langle \widehat{q}_t - q^{\pi^C, \widetilde{P}_t}, \ell'_t\rangle\right]}_{\text{OMDREG}}$$

$$+ \underbrace{\mathbb{E}\left[\sum_{t=1}^{T}\langle q^{\pi^C, \widetilde{P}_t} - q^{\pi^C, P}, \ell'_t\rangle\right]}_{\text{ERROR}_1} + \underbrace{\mathbb{E}\left[\sum_{t=1}^{T}\langle q^{\pi_t, P} - \widehat{q}_t, \ell'_t\rangle\right]}_{\text{ERROR}_2},$$

(18)

where $\ell'_t$'s denote *surrogate* loss functions obtained by shifting true losses $\ell_t$'s by $-1$ (which essentially means a conversion to rewards as in Remark 2). A new term $\text{ERROR}_1$ is the gap between an occupancy (induced by $\pi^C$) on $\widetilde{P}_t$ and its counterpart on $P$. While the comparator in OMDREG becomes $q^{\pi^C, \widetilde{P}_t}$, we can still bound it by $\mathcal{C}(\pi^C)$ (i.e., in terms of *true transition P*). $\text{ERROR}_1$ also enjoys the $\mathcal{C}(\pi^C)$-type bound since it is directly related to $\pi^C$.

For $\text{ERROR}_2$, the confidence set-based approach in Sec. 4.3.1 yields only an undesired $\mathcal{C}_{\text{all}}$ dependence. Now, we rewrite it as $\text{ERROR}_2 = \mathbb{E}\left[\sum_{t=1}^{T}\langle q^{\pi_t, P} - q^{\pi_t, \widetilde{P}_t}, \ell_t\rangle\right]$, which, by the *biased property* of $\widetilde{P}_t$, is *non-positive*. Therefore, all three terms are bounded by either $\mathcal{C}(\pi^C)$ or zero, which implies the desired *single-policy concentrability*.

After introducing main ideas, we present the full algorithm and its theoretical guarantee. The full pseudo-code is given in Algorithm 4, which runs in epochs: 1) within each epoch $e$, it updates over all occupancies induced by a single transition function $\widetilde{P}_e$; 2) a new epoch starts whenever the number of visits to any state-action pair at any layer is doubled; 3) when each new epoch starts, the biased transition $\widetilde{P}_e$ for each state-action pair $(s, a)$ at layer $h$ is updated as

$$\widetilde{P}_e^h(s' \mid s, a) = \left(\overline{P}_e^h(s' \mid s, a) - \phi_e^h(s, a, s')\right)^+, \forall s' \in \mathcal{S},$$

(19)

$$\widetilde{P}_e^h(s_{\text{ter}} \mid s, a) = \sum_{s' \in \mathcal{S}} \min\left\{\overline{P}_e^h(s' \mid s, a), \phi_e^h(s, a, s')\right\},$$

(20)

where $\overline{P}_e^h$ is the empirical transition, $x^+ := \max\{0, x\}$, and the width of confidence interval $\phi_e^h(s, a, s')$ equals

$$\min\left\{1, 2\sqrt{\frac{\overline{P}_e^h(s' \mid s, a)\log(4THSA/\zeta)}{\max\{1, N_e^h(s, a)\}}} + \frac{14\log(4THSA/\zeta)}{3\max\{1, N_e^h(s, a)\}}\right\}.$$

(21)

Finally, a minor modification in the loss estimate is that, since $q^B = q^{\pi^B, P}$ is unknown, it is replaced with its upper confidence bound as in Gabbianelli et al. (2023, Section 3.2).

*Remark 6.* Intuitively, $\widetilde{P}_e$ pushes the learner to skip the remaining steps and jump directly to the terminating state.

---

**Algorithm 4** Off-Policy Learning with Unknown Transitions

1: **Input:** learning rate $\eta > 0$
2: **Define:** $\overline{q}_{t,h}^B(s, a) = \sum_{t'=1}^{t-1} 1_{t',h}^{\text{off}}(s, a)/t$; $\overline{P}_e^h(s' \mid s, a) = \frac{N_e^h(s, a, s')}{N_e^h(s, a)}$; $\zeta = 1/T$; $\eta_t \equiv \eta, \forall t \in [T]$
3: **Initialization:** epoch index $e = 1$; arbitrary valid transition function $\widetilde{P}_1$; uniform occupancy $\widehat{q}_1, \overline{q}_{1,h}^B$; $N_0^h(\cdot, \cdot) = N_0^h(\cdot, \cdot, \cdot) = N_1^h(\cdot, \cdot) = N_1^h(\cdot, \cdot, \cdot) = 0$
4: **for** $t = 1 : T$ **do**
5:   Construct $\widehat{\ell}_{t,h}(s, a) = \frac{1_{t,h}^{\text{off}}(s,a)\cdot(\ell_{t,h}(s,a)-1)}{\overline{q}_{t,h}^B(s,a)+\frac{\eta}{2}+\sqrt{\frac{\log(2HSAT/\zeta)}{t}}}$
6:   Calculate $\widehat{q}_{t+1} = \underset{q \in \text{OM}(\widetilde{P}_e)}{\arg\min}\left(\langle q, \widehat{\ell}_t\rangle + D_{\psi_t^{\text{SE}}}(q, \widehat{q}_t)\right)$
7:   Calculate policy $\pi_{t+1}$ induced by occupancy $\widehat{q}_{t+1}$
8:   Update visit counts as $N_e^h(\cdot, \cdot) = N_e^h(\cdot, \cdot)+1_{t,h}^{\text{off}}(\cdot, \cdot)$, $N_e^h(\cdot, \cdot, \cdot) = N_e^h(\cdot, \cdot, \cdot) + 1_{t,h}^{\text{off}}(\cdot, \cdot, \cdot)$
9:   **if** $\exists(h, s, a) \in [H] \times \mathcal{S} \times \mathcal{A}$ such that $N_e^h(s, a) \geqslant \max\{1, 2 \cdot N_{e-1}^h(s, a)\}$ **then**
10:     Update epoch index $e = e + 1$
11:     $N_e^h(\cdot, \cdot) = N_{e-1}^h(\cdot, \cdot), N_e^h(\cdot, \cdot, \cdot) = N_{e-1}^h(\cdot, \cdot, \cdot)$
12:     Calculate $\widetilde{P}_e$ as in Eqs. (19) and (20)
13:     Reset $\widehat{q}_t$ as uniform occupancy
14:   **end if**
15: **end for**

---

Via our loss-to-reward conversion, $\widetilde{P}_e$ essentially brings *pessimism* rather than optimism as in Jin et al. (2023). In fact, we would like to point out that, as some attentive readers may have already noticed, both the biased loss estimator and the biased transition function are *optimistic* in the *loss* view where they were originally proposed, but become *pessimistic* in the *reward* view, which is essentially another separation between these two views.

We formally state the regret guarantee provided by Algorithm 4 in Theorem 5 below, which shows that single-policy concentrability is achieved even with unknown transition, resolving an open question raised in Bacchiocchi et al. (2024).

**Theorem 5.** *With off-policy feedback, Algorithm 4 ensures that* $\mathbb{E}\left[\sum_{t=1}^{T}\langle q^{\pi_t, P} - q^{\pi^C, P}, \ell_t\rangle\right]$ *enjoys universal bound*

$$\widetilde{O}\left(\frac{H^2SA}{\eta} + \eta\mathcal{C}(\pi^C)T + \mathcal{C}(\pi^C)\sqrt{T} + H^{3/2}\sqrt{ST\mathcal{C}(\pi^C)}\right).$$

(22)

*Remark 7.* The first term in the regret bound above, $\frac{H^2SA}{\eta}$, is worse than its counterpart in the known-transition case by a factor of $HSA$. This is because we upper bound the number of epochs by $\widetilde{O}(HSA)$, and each epoch corresponds to an OMD restart that contributes $\widetilde{O}\left(\frac{H}{\eta}\right)$ to the regret.

*Remark 8.* Our unknown-transition analysis already captures the effect of unknown $P$: when $P$ is unknown, $q^B$ must be estimated anyway to construct the loss estimator,

regardless of $\pi^{\mathrm{B}}$ being known or not. In terms of guarantees, unknown $\pi^{\mathrm{B}}$ introduces an additional additive regret of order $\widetilde{O}(\mathcal{C}(\pi^{\mathrm{C}})\sqrt{T})$ in both MABs and MDPs.

### 4.3.3. HYBRID FEEDBACK

Settling the off-policy setting, we can combine both types of feedback via CORRAL. Similarly, we combine one on-policy base algorithm and $\Theta(\log(HT))$ off-policy ones. For the on-policy one, we use UOB-Log-barrier Policy Search from Dann et al. (2023, Alg. 13) with the state-of-the-art $\widetilde{O}(H^2 S \sqrt{AT})$ regret with on-policy feedback alone. The regret guarantee with hybrid feedback is given below.

**Theorem 6.** *When the transition is unknown, with hybrid feedback, there exists an algorithm that ensures*

$$\mathbb{E}[R(\pi^{\mathrm{C}})] = \widetilde{O}\bigg(\sqrt{T} \cdot \min\Big\{H^3 S^2 A, H\sqrt{SA\mathcal{C}(\pi^{\mathrm{C}})}$$

$$+ \mathcal{C}(\pi^{\mathrm{C}}) + H^{3/2}\sqrt{S\mathcal{C}(\pi^{\mathrm{C}})}\Big\}\bigg), \forall \pi^{\mathrm{C}} \in \Pi. \qquad (23)$$

Regardless of $\pi^{\mathrm{C}}$, we always secure a $\widetilde{O}(H^3 S^2 A \sqrt{T})$ regret. In particular, when $\mathcal{C}(\pi^{\mathrm{C}})$ is as small as $\Theta(HS)$, our bound becomes $\widetilde{O}(\sqrt{T} \cdot \min\{H^{1.5} S \sqrt{A}, H^2 S\})$, strictly better than the best on-policy bound $\widetilde{O}(H^2 S \sqrt{AT})$.

## 5. Conclusion

In this work, we studied hybrid RL with adversarial losses. Our new framework combines off-policy and on-policy feedback to achieve a $\mathcal{C}(\pi^{\mathrm{C}})$-dependent regret bound, which improves upon the best existing on-policy guarantee when the coverage is good, while maintaining a universal $\widetilde{O}(\mathrm{poly}(H, S, A)\sqrt{T})$ regret against any $\pi^{\mathrm{C}}$ even when $\mathcal{C}(\pi^{\mathrm{C}})$ is arbitrarily large. Notably, we provide the first algorithm that achieves single-policy concentrability in adversarial MDPs with unknown transitions and off-policy feedback, thereby resolving an open problem posed in Gabbianelli et al. (2023); Bacchiocchi et al. (2024).

Lastly, we point out some future directions:

- A key open question is to derive the tight frontier in hybrid learning: in the simplest MAB setting, can one improve the upper bound to $\widetilde{O}(\sqrt{T \cdot \min\{A, \mathcal{C}(\pi^{\mathrm{C}})\}})$, implying no inherent cost for model selection?

- Tight regret bounds in the off-policy setting remain open (especially when the transition is unknown).

- Another extension is to consider preference feedback (Wang et al., 2023; Tsuchiya et al., 2025) instead of absolute reward/loss signals in this work.

- Given the computational efficiency of policy optimization (PO) in learning MDPs (Luo et al., 2021; Shani

et al., 2020; Cai et al., 2020; Lancewicki et al., 2023), an important direction is to investigate its applicability to off-policy settings.

- Finally, a challenging yet important extension is to study function approximation (Jin et al., 2020b; Liu et al., 2024; 2026) as this work largely focuses on the tabular setting.

## Acknowledgments

We thank the anonymous reviewers for their insightful feedback. This work is supported in part by the National Science Foundation under Grants CNS-2312833, CAREER-2441519, CNS-2312835, and CNS-2153220.

## Impact Statement

This paper presents work whose goal is to advance the field of theoretical reinforcement learning. There are many potential societal consequences of our work, none of which we feel must be specifically highlighted here.

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

# A. Notation Table

Table 2 summarizes the main notation used throughout the paper. For central objects, we include their formal definitions. Unless otherwise specified, MAB notation is recovered from the MDP notation by taking $H = 1$ and a singleton state space.

*Table 2. Summary of notation and definitions.*

| Notation | Definition / Meaning |
|---|---|
| $T$ | Number of rounds in MABs or episodes in MDPs. |
| $H$ | Horizon length of an episodic MDP. |
| $\mathcal{A}, A$ | Action space and its cardinality $A := |\mathcal{A}|$. |
| $\mathcal{S}, S$ | State space and its cardinality $S := |\mathcal{S}|$. |
| $\Delta(\mathcal{X})$ | Probability simplex over $\mathcal{X}$: $\Delta(\mathcal{X}) := \{p \in \mathbb{R}_+^{|\mathcal{X}|} : \sum_{x \in \mathcal{X}} p(x) = 1\}$. |
| $\Pi$ | Set of feasible randomized policies; in MDPs, each $\pi \in \Pi$ satisfies $\sum_{a \in \mathcal{A}} \pi_h(a \mid s) = 1$ for all $(h, s)$. |
| $\pi_t$ | Learner's policy in round/episode $t$. |
| $\pi^{\mathrm{B}}$ | Fixed behavior policy that generates off-policy feedback. |
| $\pi^{\mathrm{C}}$ | Comparator policy. |
| $\pi^*$ | Optimal fixed policy in hindsight, e.g., $\pi^* \in \operatorname{argmin}_{\pi \in \Pi} \sum_{t=1}^T V^{\pi, P}(s_{\mathrm{init}}; \ell_t)$ in MDPs. |
| $i_t$ | The index of base algorithm sampled by the meta algorithm in the hybrid setting. |
| $a_t, a_t^{\mathrm{B}}$ | On-policy action and behavior-policy action in MABs. |
| $s_{t,h}, a_{t,h}$ | State and action at step $h$ of episode $t$ under the learner's policy in MDPs. |
| $s_{t,h}^{\mathrm{B}}, a_{t,h}^{\mathrm{B}}$ | State and action at step $h$ of episode $t$ under the behavior policy in MDPs. |
| $P$ | True transition function of the underlying MDPs. |
| $P^h(s' \mid s, a)$ | True transition probability from state $s$ to $s'$ at step $h$ when action $a$ is taken. |
| $\overline{P}_e^h$ | Empirical transition estimate at epoch $e$: $\overline{P}_e^h(s' \mid s, a) := N_e^h(s, a, s')/N_e^h(s, a)$ when $N_e^h(s, a) > 0$. |
| $\tilde{P}_e^h$ | Biased transition model used at epoch $e$ for unknown-transition off-policy learning. |
| $\phi_e^h(s, a, s')$ | Confidence width for transition estimate $\overline{P}_e^h(s' \mid s, a)$. |
| $\ell_t(a)$ | MAB loss of action $a$ in round $t$, with $\ell_t : \mathcal{A} \to [0, 1]$. |
| $\ell_{t,h}(s, a)$ | MDP loss of taking action $a$ in state $s$ at step $h$ of episode $t$, with $\ell_{t,h} : \mathcal{S} \times \mathcal{A} \to [0, 1]$. |
| $V^{\pi, P}(s_{\mathrm{init}}; \ell_t)$ | Expected episode loss: $V^{\pi, P}(s_{\mathrm{init}}; \ell_t) := \mathbb{E}[\sum_{h=1}^H \ell_{t,h}(s_{t,h}, a_{t,h}) \mid P, \pi]$. |
| $R(\pi^{\mathrm{C}})$ | Regret against $\pi^{\mathrm{C}}$; in MDPs, $\mathbb{E}[R(\pi^{\mathrm{C}})] := \mathbb{E}[\sum_{t=1}^T (V^{\pi_t, P}(s_{\mathrm{init}}; \ell_t) - V^{\pi^{\mathrm{C}}, P}(s_{\mathrm{init}}; \ell_t))]$. |
| $q^{\pi, P'}$ | Occupancy measure induced by policy $\pi$ and transition $P'$. |
| $q_h^{\pi, P'}(s, a)$ | State-action occupancy probability at state $s$ and action $a$. |
| $q_t$ | Occupancy measure $q^{\pi_t, P}$ induced by the learner's policy $\pi_t$ and true transition $P$. |
| $q^{\mathrm{B}}$ | Occupancy measure $q^{\mathrm{B}} := q^{\pi^{\mathrm{B}}, P}$ induced by the behavior policy $\pi^{\mathrm{B}}$. |
| $q^{\mathrm{C}}$ | Occupancy measure $q^{\mathrm{C}} := q^{\pi^{\mathrm{C}}, P}$ induced by the comparator policy $\pi^{\mathrm{C}}$. |
| $\widehat{q}_t$ | Occupancy iterate maintained by online learning algorithms (e.g., OMD). |
| $\mathrm{OM}(P')$ | Occupancy measure set under transition $P'$: $\mathrm{OM}(P') := \{q^{\pi, P'} : \pi \in \Pi\}$. |
| $\mathrm{OM}(\mathcal{P}')$ | Occupancy measure set over transition set $\mathcal{P}'$: $\mathrm{OM}(\mathcal{P}') := \{q^{\pi, P'} : P' \in \mathcal{P}', \pi \in \Pi\}$. |
| $\mathcal{C}(\pi)$ | Coverage ratio. In MABs, $\mathcal{C}(\pi) := \sum_{a \in \mathcal{A}} \pi(a)/\pi^{\mathrm{B}}(a)$; in MDPs, $\mathcal{C}(\pi) := \sum_{h,s,a} q_h^{\pi, P}(s, a)/q_h^{\mathrm{B}}(s, a)$. |
| $C_{\mathrm{all}}$ | Worst-case coverage ratio: $C_{\mathrm{all}} := \sup_{\pi \in \Pi} \mathcal{C}(\pi)$. |
| $1_{t,h}^{\mathrm{on}}, 1_{t,h}^{\mathrm{off}}$ | Visit indicators. |
| $N_e^h(s, a), N_e^h(s, a, s')$ | Visit counts for state-action pairs and state-action-state triples during the $e$-th epoch. |
| $\psi_t^{\mathrm{SE}}, \psi_t^{\mathrm{LB}}$ | Shannon-entropy and log-barrier regularizers used in OMD. |
| $D_\psi(u, w)$ | Bregman divergence: $D_\psi(u, w) := \psi(u) - \psi(w) - \langle \nabla\psi(w), u - w \rangle$. |

# B. Comprehensive Discussion of Related Work

**On-Policy RL.** Much existing RL work has been dedicated to study statistical limits (in terms of sample complexity or regret) with on-policy feedback, which is also called bandit feedback, as learner observes the outcome associated with the executed policy. In this setup, by strategic and adaptive exploration (e.g., the well-known optimism principle), one can ensure provable convergence to the optimal policy. We give main references for both stochastic MDPs (Azar et al., 2017; Jin et al., 2018; Domingues et al., 2021; Zhang et al., 2025) and adversarial MDPs (Zimin & Neu, 2013; Jin et al., 2020a; Luo et al., 2021; Dai et al., 2022).

**Off-Policy RL.** Off-policy learning is mainly investigated in the stochastic setting (the distribution of transition dynamics and losses stays fixed), and off-policy data naturally come in the form of an offline dataset collected by the behavior policy and is hence often called *offline RL* (a.k.a. *batch* RL). It is of particular interest to investigate under what coverage condition (i.e., "quality" of offline dataset) is necessary for learning a good policy (Yin et al., 2021; Jin et al., 2021b; Zhan et al., 2022; Foster et al., 2022; Li et al., 2024).

In the stochastic setting, earlier works (Liu et al., 2020; Riedmiller, 2005) propose algorithms that either do not incorporate pessimism at all (i.e., they act greedily with respect to empirical estimates) or do not incorporate it strongly enough. As a result, their theoretical guarantees can be suboptimal and/or require stronger coverage assumptions. To this end, one plausible way to implement the pessimism principle is to subtract a penalty term, rather than add an exploration bonus, to the empirical estimates. This penalizes actions (or state-action pairs) with greater uncertainty more heavily, rather than encouraging them. Put differently, this can be viewed as flipping the sign of the bonus term in the canonical UCB (upper confidence bound) framework, thereby effectively constructing an LCB (lower confidence bound) index (Jin et al., 2021b).With a suitably designed Bernstein-type negative bonus term, the PEVI-Adv algorithm proposed in Xie et al. (2021) achieves the minimax-optimal sample complexity for episodic tabular MDPs, improving upon the result in Rashidinejad et al. (2021), which uses a Hoeffding-type bonus. More broadly, Li et al. (2024) introduced a refined notion of the (single-policy) concentrability coefficient for both finite- and infinite-horizon tabular MDPs, still within the LCB-based framework.

In the adversarial setting, Gabbianelli et al. (2023) initiate the off-policy learning problem in bandits, in which during online interactions, the learner does not observe any feedback associated with her executed policy, but instead observes the ones from another fixed behavior policy. Bacchiocchi et al. (2024) establish similar results in MDPs with known transition, *leaving the unknown transition scenario open*.

**Hybrid RL.** Xie et al. (2021) initiate the study of hybrid RL in stochastic tabular MDPs, in which they propose and study a setup called *policy finetuning*, in which the learner can still perform the standard on-policy online interactions, *additionally given* a behavior policy. With the goal of minimizing total sample complexity (from both online interactions and behavior policy), they show that the use of behavior policy can be reduced to a pre-collected offline dataset from it in stochastic environments. After this work, hybrid RL gets extensively studied (Wagenmaker & Pacchiano, 2023; Li et al., 2023; Tan et al., 2024; Song et al.; Huang et al., 2025), but is limited to the stochastic regime.

**The Best of Both Worlds and Learning-Augmented Algorithms.** Our result also fits into the general objective of developing algorithms that retain certain worst-case guarantees and simultaneously provide stronger performance on "favorable" instances. One closely related line of work is the best-of-both-worlds paradigm in online learning, which seeks to design a single algorithm that is robust under adversarial environments while simultaneously achieving sharper guarantees in easier stochastic regimes (Bubeck & Slivkins, 2012; Zimmert & Seldin, 2021; Jin et al., 2021a; Dann et al., 2023). Learning-augmented algorithms (a.k.a. algorithms with predictions) study how to design algorithms that can exploit (machine-learned) predictions while still maintaining provable worst-case guarantees when those predictions are wrong or not helpful (Lykouris & Vassilvitskii, 2021; Purohit et al., 2018; Wei & Zhang, 2020).

## C. MAB

### C.1. MAB (On-Policy)

*Proof of Theorem 2.* Theorem 2 (Algorithm 2 resp.) is a special case of Dann et al. (2023, Lemma 10) (Algorithm 10 therein resp.) when the policy class contains $A$ policies, and the $i$-th policy chooses the $i$-th action with probability one. In this case, we have

$$\mathbb{E}[R(\pi^{\mathrm{C}})] \leqslant \mathbb{E}[R(\pi^*)] \leqslant 2\sqrt{\frac{AT \log A}{\min_{t \in [T]} p_{t,0}}}, \forall \pi^{\mathrm{C}} \in \Pi. \tag{24}$$

$\square$

### C.2. MAB (Off-Policy)

*Proof of Theorem 1.* While our result looks identical to the one in Gabbianelli et al. (2023) (which take the reward view), we re-prove it here from the loss view.

For any $t \in [T]$ and $a \in \mathcal{A}$, define

$$\widetilde{\ell}_t(a) := 1\{a_t^{\mathrm{B}} = a\} \frac{\ell_t(a) - 1}{\pi^{\mathrm{B}}(a)}, \tag{25}$$

which is a conditional unbiased estimator for $(\ell_t(a) - 1)$ because

$$\mathbb{E}\left[\widetilde{\ell}_t(a) \mid \mathcal{F}_{t-1}\right] = \mathbb{E}\left[1\{a_t^{\mathrm{B}} = a\} \frac{\ell_t(a) - 1}{\pi^{\mathrm{B}}(a)} \mid \mathcal{F}_{t-1}\right] = \frac{\ell_t(a) - 1}{\pi^{\mathrm{B}}(a)} \mathbb{E}\left[1\{a_t^{\mathrm{B}} = a\} \mid \mathcal{F}_{t-1}\right] = \ell_t(a) - 1, \tag{26}$$

where $\mathcal{F}_{t-1} := \{a_1^{\mathrm{B}}, \ell_1(a_1^{\mathrm{B}}), \ldots, a_{t-1}^{\mathrm{B}}, \ell_{t-1}(a_{t-1}^{\mathrm{B}})\}$

By the regret definition, we have

$$\mathbb{E}[R(\pi^C)] = \mathbb{E}\left[\sum_{t=1}^{T} \langle \pi_t - \pi^{\mathrm{C}}, \ell_t \rangle\right] = \mathbb{E}\left[\sum_{t=1}^{T} \langle \pi_t - \pi^{\mathrm{C}}, \ell_t - \mathbf{1} \rangle\right] = \mathbb{E}\left[\sum_{t=1}^{T} \langle \pi_t - \pi^{\mathrm{C}}, \widetilde{\ell}_t \rangle\right], \tag{27}$$

where $\mathbf{1}$ denotes the all-one vector with proper dimensions throughput the this paper. Then, we further decompose the regret into two parts as

$$\mathbb{E}[R(\pi^C)] = \mathbb{E}\left[\sum_{t=1}^{T} \langle \pi_t - \pi^{\mathrm{C}}, \widehat{\ell}_t \rangle\right] + \mathbb{E}\left[\sum_{t=1}^{T} \langle \pi_t - \pi^{\mathrm{C}}, \widetilde{\ell}_t - \widehat{\ell}_t \rangle\right] \tag{28}$$

and bound these two parts respectively.

For the first part, we have

$$\mathbb{E}\left[\sum_{t=1}^{T} \langle \pi_t - \pi^{\mathrm{C}}, \widehat{\ell}_t \rangle\right]$$

$$\leqslant \frac{\log A}{\eta} + \frac{1}{\eta} \mathbb{E}\left[\sum_{t=1}^{T} \sum_{a \in \mathcal{A}} \pi_t(a) \left(\eta \widehat{\ell}_t(a) - 1 + \exp(-\eta \widehat{\ell}_t(a))\right)\right]$$

$$= \frac{\log A}{\eta} + \frac{1}{\eta} \mathbb{E}\left[\sum_{t=1}^{T} \sum_{a \in \mathcal{A}} \pi_t(a) \left(\eta \widehat{\ell}_t(a) - 1 + \exp\left(\eta \frac{1\{a_t^{\mathrm{B}} = a\}(1 - \ell_t(a))}{\pi^{\mathrm{B}}(a) + \frac{\eta}{2}}\right)\right)\right]$$

$$\leqslant \frac{\log A}{\eta} + \frac{1}{\eta} \mathbb{E}\left[\sum_{t=1}^{T} \sum_{a \in \mathcal{A}} \pi_t(a) \left(\eta \widehat{\ell}_t(a) - 1 + \exp\left(\frac{\eta}{2 \cdot \eta/2} \log\left(1 + \eta \frac{1\{a_t^{\mathrm{B}} = a\}(1 - \ell_t(a))}{\pi^{\mathrm{B}}(a)}\right)\right)\right)\right]$$

$$= \frac{\log A}{\eta} + \frac{1}{\eta} \mathbb{E}\left[\sum_{t=1}^{T} \sum_{a \in \mathcal{A}} \pi_t(a) \left(\eta \widehat{\ell}_t(a) - 1 + 1 + \eta \frac{1\{a_t^{\mathrm{B}} = a\}(1 - \ell_t(a))}{\pi^{\mathrm{B}}(a)}\right)\right]$$

$$= \frac{\log A}{\eta} + \mathbb{E}\left[\sum_{t=1}^{T} \sum_{a \in \mathcal{A}} \pi_t(a) \left(\widehat{\ell}_t(a) + \frac{1\{a_t^{\mathrm{B}} = a\}(1 - \ell_t(a))}{\pi^{\mathrm{B}}(a)}\right)\right]$$

$$= \frac{\log A}{\eta} + \mathbb{E}\left[\sum_{t=1}^{T} \langle \pi_t, \widehat{\ell}_t - \widetilde{\ell}_t \rangle\right], \tag{29}$$

where the first inequality is from Lemma 1, and the second inequality is from Lemma 7.

Therefore, we have

$$\mathbb{E}[R(\pi^C)] = \mathbb{E}\left[\sum_{t=1}^{T} \langle \pi_t - \pi^{\mathrm{C}}, \widehat{\ell}_t \rangle\right] + \mathbb{E}\left[\sum_{t=1}^{T} \langle \pi_t - \pi^{\mathrm{C}}, \widetilde{\ell}_t - \widehat{\ell}_t \rangle\right]$$

$$\leqslant \frac{\log A}{\eta} + \mathbb{E}\left[\sum_{t=1}^{T} \langle \pi_t, \widehat{\ell}_t - \widetilde{\ell}_t \rangle\right] + \mathbb{E}\left[\sum_{t=1}^{T} \langle \pi_t - \pi^{\mathrm{C}}, \widetilde{\ell}_t - \widehat{\ell}_t \rangle\right]$$

$$= \frac{\log A}{\eta} + \mathbb{E}\left[\sum_{t=1}^{T} \langle \pi^{\mathrm{C}}, \widehat{\ell}_t - \widetilde{\ell}_t \rangle\right]. \tag{30}$$

We complete the proof by arriving at

$$
\begin{aligned}
\mathbb{E}\left[\sum_{t=1}^{T}\langle \pi^{\mathrm{C}}, \widehat{\ell}_t - \widetilde{\ell}_t\rangle\right] &= \mathbb{E}\left[\sum_{t=1}^{T}\sum_{a\in\mathcal{A}}\left(\pi^{\mathrm{C}}(a)\cdot\left(1\{a_t^{\mathrm{B}}=a\}\frac{\ell_t(a)-1}{\pi^{\mathrm{B}}(a)+\eta/2} - 1\{a_t^{\mathrm{B}}=a\}\frac{\ell_t(a)-1}{\pi^{\mathrm{B}}(a)}\right)\right)\right]\\
&= \mathbb{E}\left[\sum_{t=1}^{T}\sum_{a\in\mathcal{A}}\left(\pi^{\mathrm{C}}(a)\cdot(1-\ell_t(a))\cdot 1\{a_t^{\mathrm{B}}=a\}\left(\frac{1}{\pi^{\mathrm{B}}(a)} - \frac{1}{\pi^{\mathrm{B}}(a)+\eta/2}\right)\right)\right]\\
&\leqslant \sum_{t=1}^{T}\sum_{a\in\mathcal{A}}\left(\pi^{\mathrm{C}}(a)\cdot\pi^{\mathrm{B}}(a)\frac{\eta/2}{\pi^{\mathrm{B}}(a)(\pi^{\mathrm{B}}(a)+\eta/2)}\right)\\
&\leqslant \eta/2\cdot\sum_{t=1}^{T}\sum_{a\in\mathcal{A}}\frac{\pi^{\mathrm{C}}(a)}{\pi^{\mathrm{B}}(a)} = \frac{1}{2}T\mathcal{C}(\pi^{\mathrm{C}}).
\end{aligned}
\tag{31}
$$

$\square$

*Proof of Theorem 3.* The $i$-th (off-policy) base algorithm ($i \geqslant 1$) has learning rate $\eta^i = \frac{1}{\sqrt{\exp(i)T}}$.

For any $\mathcal{C}(\pi^{\mathrm{C}}) \in [1, T]$, there must exist some base algorithm $i'$ such that

$$
\frac{1}{\sqrt{\exp(1)\mathcal{C}(\pi^{\mathrm{C}})\,T}} \leqslant \eta^{i'} \leqslant \frac{1}{\sqrt{\mathcal{C}(\pi^{\mathrm{C}})T}}.
\tag{32}
$$

Then, for the specific $\pi^{\mathrm{C}}$ and the corresponding $i'$, we have

$$
\mathbb{E}[R(\pi^C)] = \widetilde{O}\left(\frac{1}{\eta^{i'}} + \eta^{i'}T\mathcal{C}(\pi^{\mathrm{C}})\right) = \widetilde{O}\left(\sqrt{\mathcal{C}(\pi^{\mathrm{C}})T}\right).
\tag{33}
$$

$\square$

## C.3. MAB (Hybrid)

*Proof of Theorem 4.* Recall that under the CORRAL framework, we create $O(\log T)$ base instances. By Lemma 10, we have

$$
\mathbb{E}\left[\sum_{t=1}^{T}\langle p_t - u_i, \widehat{c}_t\rangle\right] = \widetilde{O}\left(\frac{1}{\eta^{\mathrm{meta}}} + \eta^{\mathrm{meta}}T - \frac{\mathbb{E}\left[\rho_{T,i}\right]}{\eta^{\mathrm{meta}}}\right), \forall i \geqslant 0,
\tag{34}
$$

where $u_i$ denotes the one-hot vector with proper dimension, whose $i$-th element is one and others are all zero.

We define $c_t(i) := \langle \pi_t^i, \ell_t\rangle$, which satisfies that

$$
\begin{aligned}
\mathbb{E}\left[\widehat{c}_t(i) \mid p_t, \mathcal{F}_{t-1}\right] &= \mathbb{E}\left[\frac{1\{i_t = i\}\ell_t(a_t)}{p_{t,i}} \mid p_t, \mathcal{F}_{t-1}\right]\\
&= (1 - p_{t,i})\cdot 0 + p_{t,i}\cdot\frac{\langle\pi_t^i, \ell_t\rangle}{p_{t,i}}\\
&= \langle\pi_t^i, \ell_t\rangle = c_t(i).
\end{aligned}
\tag{35}
$$

To connect it to the regret in terms of policy, we first have

$$
\mathbb{E}\left[\sum_{t=1}^{T}\langle \pi_t^{i_t} - \pi_t^{i}, \ell_t\rangle\right] = \sum_{t=1}^{T}\mathbb{E}\left[(c_t(i_t) - c_t(i))\right]
$$
$$
= \sum_{t=1}^{T}\mathbb{E}\left[\mathbb{E}\left[c_t(i_t) - c_t(i) \mid p_t, \mathcal{F}_{t-1}\right]\right]
$$
$$
= \sum_{t=1}^{T}\mathbb{E}\left[\mathbb{E}\left[\langle p_t - u_i, c_t\rangle \mid p_t, \mathcal{F}_{t-1}\right]\right]
$$
$$
= \sum_{t=1}^{T}\mathbb{E}\left[\langle p_t - u_i, \widehat{c}_t\rangle\right]. \tag{36}
$$

In general, we have

$$
\mathbb{E}[R(\pi^{\mathrm{C}})] = \mathbb{E}\left[\sum_{t=1}^{T}\langle \pi_t^{i_t} - \pi^{\mathrm{C}}, \ell_t\rangle\right] = \mathbb{E}\left[\sum_{t=1}^{T}\langle \pi_t^{i_t} - \pi_t^{i}, \ell_t\rangle\right] + \mathbb{E}\left[\sum_{t=1}^{T}\langle \pi_t^{i} - \pi^{\mathrm{C}}, \ell_t\rangle\right], \forall i \geqslant 0. \tag{37}
$$

Therefore, for any comparator policy $\pi^{\mathrm{C}}$, we have all following bounds hold simultaneously for any learning rate $\eta > 0$ ($\rho_{T,i} \geqslant \frac{1}{\min_{t\in[T]} p_{t,i}}$ as in Algorithm 3):

$$
\mathbb{E}[R(\pi^{\mathrm{C}})] = \widetilde{O}\left(\mathbb{E}\left[\frac{1}{\eta^{\mathrm{meta}}} + \eta^{\mathrm{meta}}T - \frac{\rho_{T,0}}{\eta^{\mathrm{meta}}} + (\rho_{T,0})^{1/2}\sqrt{AT}\right]\right), \tag{38}
$$

$$
\mathbb{E}[R(\pi^{\mathrm{C}})] = \widetilde{O}\left(\frac{1}{\eta^{\mathrm{meta}}} + \eta^{\mathrm{meta}}T + \eta^{\mathrm{meta}}AT\right), \tag{39}
$$

$$
\mathbb{E}[R(\pi^{\mathrm{C}})] = \widetilde{O}\left(\frac{1}{\eta^{\mathrm{meta}}} + \eta^{\mathrm{meta}}T + \sqrt{\mathcal{C}(\pi^{\mathrm{C}})T}\right), \tag{40}
$$

where the first bound comes from the on-policy base algorithm; the second bound is implied by the first one by viewing $\left(-\frac{\rho_{T,0}}{\eta} + (\rho_{T,0})^{1/2}\sqrt{AT}\right)$ as a quadratic function of $(\rho_{T,0})^{1/2}$; the third one is from the off-policy algorithms.

Choosing $\eta^{\mathrm{meta}} = \frac{1}{\sqrt{T}}$ completes the proof of Theorem 4. $\qquad\square$

## D. MDP (Known Transition)

The off-policy results are from Bacchiocchi et al. (2024) and the on-policy results are from Zimin & Neu (2013). We give our results in the hybrid feedback setting.

**Theorem 7.** *When transition function $P$ is known, with hybrid feedback, there exists an algorithm that ensures*

$$
\mathbb{E}[R(\pi^{\mathrm{C}})] = \widetilde{O}\left(\sqrt{T} \cdot \min\{HSA, \sqrt{\mathcal{C}(\pi^{\mathrm{C}})H}\}\right), \forall \pi^{\mathrm{C}} \in \Pi. \tag{41}
$$

Still, due to hybrid feedback, we are able to preserve the $\widetilde{O}(\sqrt{\mathcal{C}(\pi^{\mathrm{C}})HT})$ bound as in the off-policy feedback setting (Bacchiocchi et al., 2024) while being universal. Moreover, we always enjoy a $\widetilde{O}(HSA\sqrt{T})$ bound regardless of $\pi^{\mathrm{C}}$, and break the $\widetilde{\Theta}(H\sqrt{SAT})$ on-policy barrier (Zimin & Neu, 2013; Domingues et al., 2021) whenever $\mathcal{C}(\pi^{\mathrm{C}}) = o(HSA)$. For example, even when $\pi^{\mathrm{C}}$ is the optimal policy $\pi^*$, $\mathcal{C}(\pi^*)$ can be as small as $\Theta(HS)$ so that $\mathbb{E}[R(\pi^*)] = \widetilde{O}(H\sqrt{ST})$, which is $\sqrt{A}$ better than the $\widetilde{\Theta}(H\sqrt{SAT})$ minimax on-policy regret.

In the remainder of this section, we prove Theorem 7. While most results have been established from existing work (or just need minor adaptation), we provide full proof for completeness. From a high level, the only difference (compared to MAB) is that the decision space becomes the occupancy measure space, since the true transition $P$ is known.

We start with decomposing the regret into two levels.

---

**Algorithm 5** Online Relative Entropy Policy Search (O-REPS) (Zimin & Neu, 2013)

1: **Define:** Learning rate $\eta_t = \sqrt{\frac{\log(SA)}{SA \sum_{t'=1}^{t} \frac{1}{p_{t',0}}}}$

2: **Initialization:** policy $\pi_{1,h}^0(a \mid s) = 1/A, \forall (h,s,a) \in [H] \times \mathcal{S} \times \mathcal{A}$; occupancy $q_1^0 = q^{\pi_1^0, P}$

3: **for** $t = 1 : T$ **do**

4:     Get $\{i_t, p_{t,0}\}$ and on-policy feedback from Alg. 7

5:     Construct $\widehat{\ell}_{t,h}(s,a) = \frac{\mathrm{upd}_t}{p_{t,0}} \cdot 1_{t,h}^{\mathrm{on}}(s,a) \frac{\ell_{t,h}(s,a)}{q_{t,h}^0(s,a)}, \forall (h,s,a) \in [H] \times \mathcal{S} \times \mathcal{A}$

6:     Calculate $q_{t+1}^0 = \underset{q \in \mathrm{OM}(P)}{\mathrm{argmin}} \; (\langle q, \widehat{\ell}_t \rangle + D_{\psi_t^{\mathrm{SE}}}(q, q_t^0))$

7:     Induce policy $\pi_{t+1}^0$ from $q_{t+1}^0$

8: **end for**

---

$$
\begin{aligned}
\mathbb{E}[R(\pi^{\mathrm{C}})] &= \mathbb{E}\left[\sum_{t=1}^{T} V^{\pi_t^{i_t}, P}(s_{\mathrm{init}}; \ell_t) - \sum_{t=1}^{T} V^{\pi^{\mathrm{C}}, P}(s_0; \ell_t)\right] \\
&= \underbrace{\mathbb{E}\left[\sum_{t=1}^{T} V^{\pi_t^{i_t}, P}(s_{\mathrm{init}}; \ell_t) - \sum_{t=1}^{T} V^{\pi_t^{i}, P}(s_{\mathrm{init}}; \ell_t)\right]}_{\mathrm{METAREG}_i} + \underbrace{\mathbb{E}\left[\sum_{t=1}^{T} V^{\pi_t^{i}, P}(s_{\mathrm{init}}; \ell_t) - \sum_{t=1}^{T} V^{\pi^{\mathrm{C}}, P}(s_{\mathrm{init}}; \ell_t)\right]}_{\mathrm{BASEREG}_i}, \forall i.
\end{aligned}
\tag{42}
$$

### D.1. MDPs with Known Transition (On-Policy)

The on-policy results are from Zimin & Neu (2013). The full pseudo-code of the on-policy base algorithm is given in Algorithm 5.

**Theorem 8.** *Algorithm 5 ensures that*

$$
\mathrm{BASEREG}_0 = \mathbb{E}\left[\sum_{t=1}^{T} \langle q_t^0 - q^{\pi^{\mathrm{C}}, P}, \ell_t \rangle\right] = \widetilde{O}\left(H\sqrt{\frac{SAT}{\min_{t \in [T]} p_{t,0}}}\right), \forall \pi^{\mathrm{C}} \in \Pi.
\tag{43}
$$

*Proof.* First, we note that $\widehat{\ell}_{t,h}(s,a) = \frac{\mathrm{upd}_t}{p_{t,0}} \cdot 1\{a = a_t\} \frac{\ell_{t,h}(s,a)}{q_{t,h}^0(s,a)}$ is an unbiased estimate for $\ell_{t,h}(s,a)$.

Then, by the standard online learning analysis (Zimin & Neu, 2013; Dann et al., 2023), we have

$$
\mathbb{E}\left[\sum_{t=1}^{T} \langle q_t^0 - q^{\pi^{\mathrm{C}}, P}, \ell_t \rangle\right] = \widetilde{O}\left(\frac{H}{\eta_T} + HSA \sum_{t=1}^{T} \frac{\eta_t}{q_{t,0}}\right) \leqslant \widetilde{O}\left(H\sqrt{SA \sum_{t=1}^{T} \frac{1}{p_{t,0}}}\right) \leqslant \widetilde{O}\left(H\sqrt{\frac{SAT}{\min_{t \in [T]} p_{t,0}}}\right).
\tag{44}
$$

$\square$

To understand this result, on top of the standard $\widetilde{O}\left(H\sqrt{SAT}\right)$ on-policy regret, we additionally suffer a $\sqrt{\frac{1}{\min_{t \in [T]} p_{t,0}}}$ factor due to corralling. This phenomenon is the same as in the MAB setup (Theorem 2).

### D.2. MDPs with Known Transition (Off-Policy)

When the transition function $P$ is known, off-policy results have been established in Bacchiocchi et al. (2024). We give all needed proof (again) here. The pseudo-code of the off-policy base algorithm is given in Algorithm 6.

**Theorem 9.** *For any fixed learning rate $\eta > 0$, Algorithm 6 ensures that*

$$
\mathbb{E}\left[\sum_{t=1}^{T} \langle q_t - q^{\pi^{\mathrm{C}}, P}, \ell_t \rangle\right] = \widetilde{O}\left(\frac{H}{\eta} + \eta \mathcal{C}(\pi^{\mathrm{C}})T\right), \forall \pi^{\mathrm{C}} \in \Pi.
\tag{45}
$$

---

**Algorithm 6** Pessimistic Relative Entropy Policy Search (PREPS) (Bacchiocchi et al., 2024)

---

1: **Input:** learning rate $\eta > 0$
2: **Initialization:** policy $\pi_{1,h}^0(a \mid s) = 1/A, \forall(h, s, a) \in [H] \times \mathcal{S} \times \mathcal{A}$; occupancy $q_1 = q^{\pi_1, P}$; $\eta_t \equiv \eta, \forall t \in [T]$
3: **for** $t = 1 : T$ **do**
4:     Observe off-policy feedback
5:     Construct loss estimate

$$\widehat{\ell}_{t,h}(s, a) = \frac{1_{t,h}^{\text{off}}(s, a) \cdot (\ell_{t,h}(s, a) - 1)}{q_h^{\text{B}}(s, a) + \frac{\eta}{2}}$$

6:     Calculate $q_{t+1} = \underset{q \in \text{OM}(P)}{\text{argmin}} \ (\langle q, \widehat{\ell}_t \rangle + D_{\psi_t^{\text{SE}}}(q, q_t))$
7:     Induce policy $\pi_{t+1}$ from $q_{t+1}$
8: **end for**

---

*Proof.* Define $\widetilde{\ell}_{t,h}(s, a) := 1_{t,h}^{\text{off}}(s, a)\frac{\ell_{t,h}(s,a)-1}{q_h^{\text{B}}(s,a)}$, which is an unbiased estimator for $(\ell_{t,h}(s, a) - 1)$. Then we have

$$
\begin{aligned}
\mathbb{E}\left[\sum_{t=1}^{T}\langle q_t - q^{\pi^{\text{C}}, P}, \ell_t\rangle\right] &= \mathbb{E}\left[\sum_{t=1}^{T}\langle q_t - q^{\pi^{\text{C}}, P}, \ell_t - \mathbf{1}\rangle\right] \\
&= \mathbb{E}\left[\sum_{t=1}^{T}\langle q_t - q^{\pi^{\text{C}}, P}, \widetilde{\ell}_t\rangle\right] \\
&= \mathbb{E}\left[\sum_{t=1}^{T}\langle q_t - q^{\pi^{\text{C}}, P}, \widehat{\ell}_t\rangle\right] + \mathbb{E}\left[\sum_{t=1}^{T}\langle q_t - q^{\pi^{\text{C}}, P}, \widetilde{\ell}_t - \widehat{\ell}_t\rangle\right].
\end{aligned} \tag{46}
$$

For the first term, we have

$$
\begin{aligned}
&\mathbb{E}\left[\sum_{t=1}^{T}\langle q^{\pi_t, P} - q^{\pi^{\text{C}}, P}, \widehat{\ell}_t\rangle\right] \\
&\leqslant \widetilde{O}\left(\frac{H}{\eta}\right) + \frac{1}{\eta}\mathbb{E}\left[\sum_{t=1}^{T}\sum_{h=1}^{H}\sum_{(s,a)\in\mathcal{S}\times\mathcal{A}} q_h^{\pi_t, P}(s, a)\left(\eta\widehat{\ell}_{t,h}(s, a) - 1 + \exp\left(-\eta\widehat{\ell}_{t,h}(s, a)\right)\right)\right] \\
&= \widetilde{O}\left(\frac{H}{\eta}\right) + \frac{1}{\eta}\mathbb{E}\left[\sum_{t=1}^{T}\sum_{h=1}^{H}\sum_{(s,a)\in\mathcal{S}\times\mathcal{A}} q_h^{\pi_t, P}(s, a)\left(\eta\widehat{\ell}_{t,h}(s, a) - 1 + \exp\left(\eta\frac{1_{t,h}^{\text{off}}(s, a)(1 - \ell_{t,h}(s, a))}{q_h^{\text{B}}(s, a) + \frac{\eta}{2}}\right)\right)\right] \\
&\leqslant \widetilde{O}\left(\frac{H}{\eta}\right) + \frac{1}{\eta}\mathbb{E}\left[\sum_{t=1}^{T}\sum_{h=1}^{H}\sum_{(s,a)\in\mathcal{S}\times\mathcal{A}} q_h^{\pi_t, P}(s, a)\left(\eta\widehat{\ell}_{t,h}(s, a) - 1 + \exp\left(\frac{\eta}{2 \cdot \eta/2}\log\left(1 + \eta\frac{1_{t,h}^{\text{off}}(s, a)(1 - \ell_{t,h}(s, a))}{q_h^{\text{B}}(s, a)}\right)\right)\right)\right] \\
&= \widetilde{O}\left(\frac{H}{\eta}\right) + \mathbb{E}\left[\sum_{t=1}^{T}\langle q^{\pi_t, P}, \widehat{\ell}_t - \widetilde{\ell}_t\rangle\right],
\end{aligned} \tag{47}
$$

where the first inequality holds due to Lemma 2; the second inequality is from Lemma 7.

Fixing an episode $t$, a layer $h$, and a state-action pair $(s, a)$, we have

$$\mathbb{E}\left[q_h^{\pi^{\mathrm{C}}, P}(s, a)\left(\widehat{\ell}_{t,h}(s, a) - \widetilde{\ell}_{t,h}(s, a)\right)\right]$$

$$= \mathbb{E}\left[q_h^{\pi^{\mathrm{C}}, P}(s, a) \cdot (1 - \ell_{t,h}(s, a)) \cdot 1_{t,h}^{\mathrm{off}}(s, a) \cdot \left(\frac{1}{q_h^{\mathrm{B}}(s, a)} - \frac{1}{q_h^{\mathrm{B}}(s, a) + \frac{\eta}{2}}\right)\right]$$

$$= \mathbb{E}\left[q_h^{\pi^{\mathrm{C}}, P}(s, a)(1 - \ell_{t,h}(s, a))\left(\frac{q_h^{\mathrm{B}}(s, a) + \frac{\eta}{2} - q_h^{\mathrm{B}}(s, a)}{q_h^{\mathrm{B}}(s, a) + \frac{\eta}{2}}\right)\right]$$

$$\leqslant \mathbb{E}\left[q_h^{\pi^{\mathrm{C}}, P}(s, a) \frac{\frac{\eta}{2}}{q_h^{\mathrm{B}}(s, a) + \frac{\eta}{2}}\right]$$

$$\leqslant \frac{\eta}{2} \cdot \frac{q_h^{\pi^{\mathrm{C}}, P}(s, a)}{q_h^{\mathrm{B}}(s, a)}, \tag{48}$$

where the first inequality is because losses are bounded in $[0, 1]$ and $\overline{q}_{t,h}^{\mathrm{B}}(s, a) + \sqrt{\frac{\log(2HSAT/\zeta)}{2t}} \geqslant q_h^{\mathrm{B}}(s, a)$.

Adding all episodes, layers, and state-action pairs together, we get

$$\mathbb{E}\left[\sum_{t=1}^{T}\langle q^{\pi^{\mathrm{C}}, P}, \widehat{\ell}_t - \widetilde{\ell}_t\rangle\right] \leqslant \frac{1}{2}\eta T \mathcal{C}(\pi^{\mathrm{C}}), \tag{49}$$

which implies that

$$\mathbb{E}\left[\sum_{t=1}^{T}\langle q^{\pi_t, P} - q^{\pi^{\mathrm{C}}, P}, \ell_t\rangle\right] = \widetilde{O}\left(\frac{H}{\eta} + \eta T \mathcal{C}(\pi^{\mathrm{C}})\right). \tag{50}$$

$\square$

Similar to the MAB case, if we create $\lfloor\log(HT) + 1\rfloor$ off-policy instances, the $i$-th one $(i \geqslant 1)$ among which has learning rate $\eta^i = \sqrt{\frac{H}{\exp(i) \cdot T}}$, then for any $\pi^{\mathrm{C}} \in \Pi$ whose $\mathcal{C}(\pi^{\mathrm{C}}) = O(HT)$, there must exist some $i \geqslant 1$ such that

$$\mathrm{BASEREG}_i = \mathbb{E}\left[\sum_{t=1}^{T}\langle q^{\pi_t^i, P} - q^{\pi^{\mathrm{C}}, P}, \ell_t\rangle\right] = \widetilde{O}\left(\sqrt{\mathcal{C}(\pi^{\mathrm{C}})HT}\right), \tag{51}$$

where $\{\pi_t^i\}_{t=1,\dots,T}$ is the policy sequence produced by the $i$-th instance $(i \geqslant 1)$.

### D.3. Meta-Level Regret

We first give the meta algorithm (Algorithm 7) for this setup. Compared to the MAB case, we replace the base algorithms, and the loss estimates should now depend on the loss in the whole trajectory.

Recall that our goal is to upper bound the meta-level regret

$$\mathrm{METAREG}_i = \mathbb{E}\left[\sum_{t=1}^{T} V^{\pi_t^{i_t}, P}(s_{\mathrm{init}}; \ell_t) - \sum_{t=1}^{T} V^{\pi_t^i, P}(s_{\mathrm{init}}; \ell_t)\right]. \tag{52}$$

In MDP, we define $c_t(i) := \langle q^{\pi_t^i, P}, \ell_t\rangle$, which satisfies that

$$\mathbb{E}\left[\widehat{c}_t(i) \mid p_t, \mathcal{F}_{t-1}\right] = \mathbb{E}\left[\frac{1\{i_t = i\}\sum_{h=1}^{H} \ell_{t,h}(s_{t,h}, a_{t,h})}{p_{t,i}} \mid p_t, \mathcal{F}_{t-1}\right]$$

$$= (1 - p_{t,i}) \cdot 0 + p_{t,i} \cdot \frac{\langle \pi_t^i, \ell_t\rangle}{p_{t,i}}$$

$$= \langle \pi_t^i, \ell_t\rangle = c_t(i). \tag{53}$$

---

**Algorithm 7** Hybrid Learning in Adversarial MDPs (Known Transition)

1: **Input:** initial learning rate $\eta^{\text{meta}} > 0$
2: **Define:** truncated probability simplex $\Omega' = \{x : \sum_i x(i) = 1 \text{ and } x(i) \geqslant \frac{H^6 S^8 A^3}{T}, \forall i\}$; factor $\kappa = \exp(1/\log T)$; number of base algorithms $N_{\text{base}} = \lfloor \log(HT) \rfloor + 2$
3: **Initialization:** $\eta_{1,i} = \eta^{\text{meta}}, \rho_{1,i} = 2N_{\text{base}}, p_{1,i} = 1/N_{\text{base}}, \forall i = 0, \ldots, N_{\text{base}} - 1$
4: **for** $t = 1 : T$ **do**
5:     Sample base algorithm (index) $i_t \sim p_t$
6:     Execute $\pi_t = \pi_t^{i_t}$ and observe on-policy feedback
7:     Pass $\{i_t, p_{t,0}\}$ and on-policy feedback to Alg. 5 and get $\pi_{t+1}^0$ from it
8:     **for** off-policy base alg. $i = 1, \ldots, N_{\text{base}} - 1$ **do**
9:         Get $\pi_{t+1}^i$ from the $i$-th instance of Alg. 6
10:     **end for**
11:     Construct $\widehat{c}_t(i) = 1\{i = i_t\} \frac{\sum_{h=1}^H \ell_{t,h}(s_{t,h}, a_{t,h})}{p_{t,i}}, \forall i$
12:     Calculate $p_{t+1} = \operatorname{argmin}_{p \in \Omega'} \left( \langle p, \widehat{c}_t \rangle + D_{\psi_t^{\text{LB}}}(p, p_t) \right)$
13:     **for** each base algorithm $i$ **do**
14:         **if** $\frac{1}{p_{t+1,i}} > \rho_{t,i}$, **then** $\rho_{t+1,i} = \frac{2}{p_{t+1,i}}, \eta_{t+1,i} = \eta_{t,i}\kappa$
15:         **else** $\rho_{t+1,i} = \rho_{t,i}, \eta_{t+1,i} = \eta_{t,i}$
16:     **end for**
17: **end for**

---

Therefore, we have

$$\mathbb{E}\left[\sum_{t=1}^T V^{\pi_t^{i_t}, P}(s_{\text{init}}; \ell_t) - \sum_{t=1}^T V^{\pi_t^i, P}(s_{\text{init}}; \ell_t)\right] = \mathbb{E}\left[\sum_{t=1}^T \langle q^{\pi_t^{i_t}, P}, \ell_t \rangle - \sum_{t=1}^T \langle q^{\pi_t^i, P}, \ell_t \rangle\right]$$

$$= \sum_{t=1}^T \mathbb{E}\left[(c_t(i_t) - c_t(i))\right]$$

$$= \sum_{t=1}^T \mathbb{E}\left[\mathbb{E}\left[c_t(i_t) - c_t(i) \mid p_t, \mathcal{F}_{t-1}\right]\right]$$

$$= \sum_{t=1}^T \mathbb{E}\left[\mathbb{E}\left[\langle p_t - u_i, c_t \rangle \mid p_t, \mathcal{F}_{t-1}\right]\right]$$

$$= \sum_{t=1}^T \mathbb{E}\left[\langle p_t - u_i, \widehat{c}_t \rangle\right]$$

$$= \sum_{t=1}^T \mathbb{E}\left[\langle p_t - u_i, c_t \rangle\right]. \tag{54}$$

By Lemma 10, we have

$$\text{METAREG}_i = \mathbb{E}\left[\sum_{t=1}^T \langle p_t - u_i, \widehat{c}_t \rangle\right] = \widetilde{O}\left(\frac{1}{\eta} + \eta H^2 T + H\sqrt{T} - \frac{\mathbb{E}[\rho_{T,i}]}{\eta}\right), \forall i \geqslant 0. \tag{55}$$

---

**Algorithm 8** Hybrid Learning in Adversarial MDP (Unknown Transition)

1: **Input:** initial learning rate $\eta^{\mathrm{meta}} > 0$
2: **Define:** truncated probability simplex $\Omega' = \{x : \sum_i x(i) = 1 \text{ and } x(i) \geqslant \frac{H^6 S^8 A^3}{T}, \forall i\}$; factor $\kappa = \exp(1/\log T)$;
    number of base algorithms $N_{\mathrm{base}} = \lfloor \log(HT) \rfloor + 2$
3: **Initialization:** $\eta_{1,i} = \eta^{\mathrm{meta}}, \rho_{1,i} = 2N_{\mathrm{base}}, p_{1,i} = 1/N_{\mathrm{base}}, \forall i = 0, \ldots, N_{\mathrm{base}} - 1$
4: **for** $t = 1 : T$ **do**
5:     Sample base algorithm (index) $i_t \sim p_t$
6:     Execute $\pi_t = \pi_t^{i_t}$ and observe on-policy feedback
7:     Pass $\{i_t, p_{t,0}\}$ and on-policy feedback to Alg. 9 and get $\pi_{t+1}^0$ from it
8:     **for** off-policy base alg. $i = 1, \ldots, N_{\mathrm{base}} - 1$ **do**
9:         Get $\pi_{t+1}^i$ from the $i$-th instance of Alg. 4
10:     **end for**
11:     Construct $\widehat{c}_t(i) = \mathbb{1}\{i = i_t\}\frac{\sum_{h=1}^{H} \ell_{t,h}(s_{t,h}, a_{t,h})}{p_{t,i}}, \forall i$
12:     Calculate $p_{t+1} = \mathrm{argmin}_{p \in \Omega'} \left( \langle p, \widehat{c}_t \rangle + D_{\psi_t^{\mathrm{LB}}}(p, p_t) \right)$
13:     **for** each base algorithm $i$ **do**
14:         **if** $\frac{1}{p_{t+1,i}} > \rho_{t,i}$, **then** $\rho_{t+1,i} = \frac{2}{p_{t+1,i}}, \eta_{t+1,i} = \eta_{t,i}\kappa$
15:         **else** $\rho_{t+1,i} = \rho_{t,i}, \eta_{t+1,i} = \eta_{t,i}$
16:     **end for**
17: **end for**

---

### D.4. Combining Meta and Base Levels Together

For any fixed learning rate $\eta^{\mathrm{meta}} > 0$ in Algorithm 7, the following bounds hold universally against all $\pi^{\mathrm{C}} \in \Pi$:

$$\mathbb{E}[R(\pi^{\mathrm{C}})] = \widetilde{O}\left( \mathbb{E}\left[ \frac{1}{\eta^{\mathrm{meta}}} + \eta^{\mathrm{meta}} H^2 T - \frac{\rho_{T,0}}{\eta^{\mathrm{meta}}} + H\sqrt{T} + (\rho_{T,0})^{1/2} H\sqrt{SAT} \right] \right), \tag{56}$$

$$\mathbb{E}[R(\pi^{\mathrm{C}})] = \widetilde{O}\left( \frac{1}{\eta^{\mathrm{meta}}} + \eta^{\mathrm{meta}} H^2 T + H\sqrt{T} + \eta^{\mathrm{meta}} H^2 SAT \right), \tag{57}$$

$$\mathbb{E}[R(\pi^{\mathrm{C}})] = \widetilde{O}\left( \frac{1}{\eta^{\mathrm{meta}}} + \eta^{\mathrm{meta}} H^2 T + H\sqrt{T} + \sqrt{\mathcal{C}(\pi^{\mathrm{C}})HT} \right). \tag{58}$$

Choosing $\eta^{\mathrm{meta}} = \frac{1}{H\sqrt{T}}$ (note that $\mathcal{C}(\pi^{\mathrm{C}}) \geqslant H$), we get

$$\mathbb{E}[R(\pi^{\mathrm{C}})] = \widetilde{O}\left( \min\{HSA, \sqrt{\mathcal{C}(\pi^{\mathrm{C}})H}\} \cdot \sqrt{T} \right), \forall \pi^{\mathrm{C}} \in \Pi. \tag{59}$$

## E. MDP (Unknown Transition)

We first give the meta algorithm for this setup in Algorithm 8.

In the remainder of this section, we prove Theorem 6. We start with decomposing the regret into two levels.

$$\mathbb{E}[R(\pi^{\mathrm{C}})] = \mathbb{E}\left[ \sum_{t=1}^{T} V^{\pi_t^{i_t}, P}(s_{\mathrm{init}}; \ell_t) - \sum_{t=1}^{T} V^{\pi^{\mathrm{C}}, P}(s_{\mathrm{init}}; \ell_t) \right]$$

$$= \underbrace{\mathbb{E}\left[ \sum_{t=1}^{T} V^{\pi_t^{i_t}, P}(s_{\mathrm{init}}; \ell_t) - \sum_{t=1}^{T} V^{\pi_t^i, P}(s_{\mathrm{init}}; \ell_t) \right]}_{\mathrm{METAREG}_i} + \underbrace{\mathbb{E}\left[ \sum_{t=1}^{T} V^{\pi_t^i, P}(s_{\mathrm{init}}; \ell_t) - \sum_{t=1}^{T} V^{\pi^{\mathrm{C}}, P}(s_{\mathrm{init}}; \ell_t) \right]}_{\mathrm{BASEREG}_i}, \forall i \geqslant 0.$$

$$\tag{60}$$

---

**Algorithm 9** UOB-Log-Barrier Policy Search (Dann et al., 2023, Algorithm 13)

---

1: **Define:** Truncated occupancy measure space $\Omega'_{\text{OM}} = \left\{ q \; : \; q(s,a,s') \geqslant \frac{1}{T^3 H^2 S^2 A} \right\}$; $\delta = \frac{1}{T^5 H^3 S^3 A}$; $\overline{P}^h_t(s' \mid s,a) = \frac{N^h_t(s,a,s')}{N^h_t(s,a)}$; $\eta_{t,i} \equiv \eta_t, \forall t, i$; COMP-UOB is Algorithm 3 of Jin et al. (2020a); $u^h_t(s,a) :=$ COMP-UOB$(\pi^0_t, h, s, a, \mathcal{P}_t), \forall (t,h,s,a) \in [T] \times [H] \times \mathcal{S} \times \mathcal{A}$

2: **Initialization:** uniform occupancy $\widehat{q}_1(s,a,s') = \frac{1}{S^2 A}$; policy $\pi^0_1$ induced by $\widehat{q}_1$; $t^* = 1$; $\eta_1 = 1$; $\mathcal{P}_1$ as the set of all valid transition functions;

3: **for** $t = 1, \ldots, T$ **do**

4:      Get $\{i_t, p_{t,0}\}$ and on-policy feedback from Algorithm 8

5:      Construct loss estimate $\widehat{\ell}_{t,h}(s,a) = \frac{\text{upd}_t}{p_{t,0}} \cdot \frac{1^{\text{on}}_{t,h}(s,a)}{u^h_t(s,a)} \cdot \ell_{t,h}(s,a)$

6:      Update counters $N^h_{t+1}(\cdot,\cdot) = N^h_t(\cdot,\cdot) + \text{upd}_t \cdot 1^{\text{on}}_{t,h}(\cdot,\cdot)$, $N^h_{t+1}(\cdot,\cdot,\cdot) = N^h_t(\cdot,\cdot,\cdot) + \text{upd}_t \cdot 1^{\text{on}}_{t,h}(\cdot,\cdot,\cdot)$

7:      Compute confidence set

$$\mathcal{P}_{t+1} = \left\{ \widehat{P} : \left| \widehat{P}(s'|s,a) - \overline{P}^h_{t+1}(s'|s,a) \right| \leqslant \phi^h_{t+1}(s'|s,a), \; \forall (s,a,s') \in \mathcal{S} \times \mathcal{A} \times \mathcal{S} \right\}$$

where

$$\phi^h_{t+1}(s'|s,a) = 4\sqrt{\frac{\overline{P}^h_{t+1}(s'|s,a) \log(HSAT/\delta)}{\max\{1, N^h_t(s,a)\}}} + \frac{28 \log(HSAT/\delta)}{3 \max\{1, N^h_t(s,a)\}}$$

8:      **if** $\sum^t_{t'=t^*} \sum_{h \in [H]} \sum_{(s,a) \in \mathcal{S} \times \mathcal{A}} \frac{\text{upd}_{t'} \cdot 1^{\text{on}}_{t',h}(s,a) \cdot (\ell_{t',h}(s,a))^2}{(p_{t',0})^2} + \max_{\tau \leq t} \frac{H}{(p_{\tau,0})^2} \geqslant \frac{H^2 S^2 A \ln(HSAT)}{(\eta_t)^2}$ **then**

9:          $\eta_{t+1} = \eta_t/2$

10:        set $\widehat{q}_{t+1}$ as uniform occupancy

11:        $t^* = t + 1$

12:      **else**

13:        $\eta_{t+1} = \eta_t$

14:        $\widehat{q}_{t+1} = \text{argmin}_{q \in \text{OM}(\mathcal{P}_{t+1}) \cap \Omega'_{\text{OM}}} \left\{ \langle q, \widehat{\ell}_t \rangle + D_{\psi^{\text{LB}}_t}(q, \widehat{q}_t) \right\}$.

15:      **end if**

16:      Induce policy $\pi^0_{t+1}$ from $\widehat{q}_{t+1}$

17: **end for**

---

### E.1. MDPs with Unknown Transitions (On-Policy)

**Theorem 10** (Lemma 45 of Dann et al. (2023)). *Algorithm 9 as the base algorithm under* CORRAL *ensures that*

$$\text{BASEREG}_0 = \widetilde{O} \left( \frac{H^2 S \sqrt{AT}}{\sqrt{\min_{t \in [T]} p_{t,0}}} + \frac{H^5 S^5 A^2}{\min_{t \in [T]} p_{t,0}} \right). \tag{61}$$

*Remark* 9. In the meta algorithm for MDPs, we truncate the probability simplex over all base algorithms so that $p_{t,i} \geqslant \frac{H^6 S^8 A^3}{T}$ and the first term $\frac{H^2 S \sqrt{AT}}{\sqrt{\min_{t \in [T]} p_{t,0}}}$ dominates the second term $\frac{H^5 S^5 A^2}{\min_{t \in [T]} p_{t,0}}$. Note that such a simplex truncation is not an additional new step due to our problem but is necessary whenever one uses log-barrier regularizer together with increasing learning rate scheduling, whose purpose is to ensure learning rates do not increase too many times. Since the number of base algorithms is $O(\log(HT))$, this only introduces an additive $O(\text{poly}(H, S, A) \cdot \log T)$ term to the total regret.

### E.2. MDPs with Unknown Transitions (Off-Policy)

In this section, we prove Theorem 5 associated with Algorithm 4.

We use $N_{\text{epoch}}$ to denote the number of total epochs. We let $e(t)$ denote the index of the epoch to which episode $t$ belongs.

We first decompose the regret as

$$
\begin{aligned}
\mathbb{E}\left[\sum_{t=1}^{T}\langle q^{\pi_t,P} - q^{\pi^C,P}, \ell_t\rangle\right] &= \mathbb{E}\left[\sum_{t=1}^{T}\langle q^{\pi_t,P} - q^{\pi^C,P}, \ell_t - \mathbf{1}\rangle\right] \\
&= \underbrace{\mathbb{E}\left[\sum_{t=1}^{T}\langle q^{\pi_t,\widetilde{P}_{e(t)}} - q^{\pi^C,\widetilde{P}_{e(t)}}, \ell_t - \mathbf{1}\rangle\right]}_{\text{OMDReg}} + \underbrace{\mathbb{E}\left[\sum_{t=1}^{T}\langle q^{\pi^C,\widetilde{P}_{e(t)}} - q^{\pi^C,P}, \ell_t - \mathbf{1}\rangle\right]}_{\text{Error}_1} \\
&\quad + \underbrace{\mathbb{E}\left[\sum_{t=1}^{T}\langle q^{\pi_t,P} - q^{\pi_t,\widetilde{P}_{e(t)}}, \ell_t - \mathbf{1}\rangle\right]}_{\text{Error}_2}.
\end{aligned}
\tag{62}
$$

### E.2.1. BOUNDING OMDREG

We still define $\widetilde{\ell}_{t,h}(s,a) := 1_{t,h}^{\text{off}}(s,a)\frac{\ell_{t,h}(s,a)-1}{q_h^{\text{B}}(s,a)}$, which is an unbiased estimator for $(\ell_{t,h}(s,a) - 1)$.

We define the following three good events (Lemmas 5, 9, and 6):

$$
P \in \cap_e \mathcal{P}_e,
\tag{63}
$$

$$
\frac{q_h^{\text{B}}(s,a)}{\max\{1, N_e^h(s,a)\}} \leqslant \frac{8\log(HSAT/\zeta)}{t_e}, \forall (h,s,a) \in [H] \times \mathcal{S} \times \mathcal{A},
\tag{64}
$$

$$
\left|\overline{q}_{t,h}^{\text{B}}(s,a) - q_h^{\text{B}}(s,a)\right| \leqslant \sqrt{\frac{\log(2HSAT/\zeta)}{2t}}, \forall (t,h,s,a) \in [T] \times [H] \times \mathcal{S} \times \mathcal{A}.
\tag{65}
$$

With probability at least $1 - \zeta/3$, all of them hold. In the remainder of the proof, we analyze the regret when they hold with $\zeta = 1/T$. With probability at most $\zeta/3$, at least one of them fails, and in this case we trivially bound the total regret by $HT$, and the contribution from this part to $\mathbb{E}[R(\pi^C)]$ is bounded by $\zeta HT/3 \leqslant H/3$.

We further decompose OMDREG as

$$
\begin{aligned}
\mathbb{E}\left[\sum_{t=1}^{T}\langle q^{\pi_t,\widetilde{P}_{e(t)}} - q^{\pi^C,\widetilde{P}_{e(t)}}, \ell_t - \mathbf{1}\rangle\right] &= \mathbb{E}\left[\sum_{t=1}^{T}\langle q^{\pi_t,\widetilde{P}_{e(t)}} - q^{\pi^C,\widetilde{P}_{e(t)}}, \widetilde{\ell}_t\rangle\right] \\
&= \mathbb{E}\left[\sum_{t=1}^{T}\langle q^{\pi_t,\widetilde{P}_{e(t)}} - q^{\pi^C,\widetilde{P}_{e(t)}}, \widehat{\ell}_t\rangle\right] + \mathbb{E}\left[\sum_{t=1}^{T}\langle q^{\pi_t,\widetilde{P}_{e(t)}} - q^{\pi^C,\widetilde{P}_{e(t)}}, \widetilde{\ell}_t - \widehat{\ell}_t\rangle\right].
\end{aligned}
\tag{66}
$$

In the first term, for each epoch $e$, we have

$$\mathbb{E}\left[\sum_{t=t_e}^{t_{e+1}-1} \langle q^{\pi_t, \widetilde{P}_e} - q^{\pi^C, \widetilde{P}_e}, \widehat{\ell}_t\rangle\right]$$

$$\leqslant \widetilde{O}\left(\frac{H}{\eta}\right) + \frac{1}{\eta}\mathbb{E}\left[\sum_{t=t_e}^{t_{e+1}-1}\sum_{h=1}^{H}\sum_{(s,a)\in\mathcal{S}\times\mathcal{A}} q_h^{\pi_t, \widetilde{P}_e}(s,a)\left(\eta\widehat{\ell}_{t,h}(s,a) - 1 + \exp\left(-\eta\widehat{\ell}_{t,h}(s,a)\right)\right)\right]$$

$$= \widetilde{O}\left(\frac{H}{\eta}\right) + \frac{1}{\eta}\mathbb{E}\left[\sum_{t=t_e}^{t_{e+1}-1}\sum_{h=1}^{H}\sum_{(s,a)\in\mathcal{S}\times\mathcal{A}} q_h^{\pi_t, \widetilde{P}_e}(s,a)\left(\eta\widehat{\ell}_{t,h}(s,a) - 1 + \exp\left(\eta\frac{1_{t,h}^{\text{off}}(s,a)(1-\ell_{t,h}(s,a))}{\overline{q}_{t,h}^{\text{B}}(s,a) + \frac{\eta}{2} + \sqrt{\frac{\log(2HSAT/\zeta)}{2t}}}\right)\right)\right]$$

$$\leqslant \widetilde{O}\left(\frac{H}{\eta}\right) + \frac{1}{\eta}\mathbb{E}\left[\sum_{t=t_e}^{t_{e+1}-1}\sum_{h=1}^{H}\sum_{(s,a)\in\mathcal{S}\times\mathcal{A}} q_h^{\pi_t, \widetilde{P}_e}(s,a)\left(\eta\widehat{\ell}_{t,h}(s,a) - 1 + \exp\left(\eta\frac{1_{t,h}^{\text{off}}(s,a)(1-\ell_{t,h}(s,a))}{q_h^{\text{B}}(s,a) + \frac{\eta}{2}}\right)\right)\right]$$

$$\leqslant \widetilde{O}\left(\frac{H}{\eta}\right) + \frac{1}{\eta}\mathbb{E}\left[\sum_{t=t_e}^{t_{e+1}-1}\sum_{h=1}^{H}\sum_{(s,a)\in\mathcal{S}\times\mathcal{A}} q_h^{\pi_t, \widetilde{P}_e}(s,a)\left(\eta\widehat{\ell}_{t,h}(s,a) - 1 + \exp\left(\frac{\eta}{2\cdot\eta/2}\log\left(1 + \eta\frac{1_{t,h}^{\text{off}}(s,a)(1-\ell_{t,h}(s,a))}{q_h^{\text{B}}(s,a)}\right)\right)\right)\right]$$

$$= \widetilde{O}\left(\frac{H}{\eta}\right) + \mathbb{E}\left[\sum_{t=t_e}^{t_{e+1}-1} \langle q^{\pi_t, \widetilde{P}_e}, \widehat{\ell}_t - \widetilde{\ell}_t\rangle\right], \tag{67}$$

where the first inequality holds due to Lemma 2; the second inequality is because $\overline{q}_{t,h}^{\text{B}}(s,a) + \sqrt{\frac{\log(2HSAT/\zeta)}{2t}} \geqslant q_h^{\text{B}}(s,a)$ due to the good event (Lemma 6); the third inequality is from Lemma 7.

Adding all epochs together, we have

$$\mathbb{E}\left[\sum_{t=1}^{T} \langle q^{\pi_t, \widetilde{P}_{e(t)}} - q^{\pi^C, \widetilde{P}_{e(t)}}, \ell_t - \mathbf{1}\rangle\right] \leqslant \widetilde{O}\left(\frac{\mathbb{E}[N_{\text{epoch}}]H}{\eta}\right) + \mathbb{E}\left[\sum_{t=1}^{T} \langle q^{\pi_t, \widetilde{P}_{e(t)}}, \widehat{\ell}_t - \widetilde{\ell}_t\rangle\right] + \mathbb{E}\left[\sum_{t=1}^{T} \langle q^{\pi_t, \widetilde{P}_{e(t)}} - q^{\pi^C, \widetilde{P}_{e(t)}}, \widetilde{\ell}_t - \widehat{\ell}_t\rangle\right]$$

$$\leqslant \widetilde{O}\left(\frac{H^2 SA}{\eta}\right) + \mathbb{E}\left[\sum_{t=1}^{T} \langle q^{\pi^C, \widetilde{P}_{e(t)}}, \widehat{\ell}_t - \widetilde{\ell}_t\rangle\right], \tag{68}$$

where in the last step, $N_{\text{epoch}}$ is bounded by $\widetilde{O}(HSA)$ due to Lemma 8. It is left to bound $\mathbb{E}\left[\sum_{t=1}^{T} \langle q^{\pi^C, \widetilde{P}_{e(t)}}, \widehat{\ell}_t - \widetilde{\ell}_t\rangle\right]$.

Fixing an episode $t$, a layer $h$, and a state-action pair $(s, a)$, we have

$$
\mathbb{E}\left[q_h^{\pi^{\mathrm{C}}, \widetilde{P}_{e(t)}}(s, a)\left(\widehat{\ell}_{t,h}(s, a) - \widetilde{\ell}_{t,j}(s, a)\right)\right]
$$

$$
= \mathbb{E}\left[q_h^{\pi^{\mathrm{C}}, \widetilde{P}_{e(t)}}(s, a) \cdot (1 - \ell_{t,h}(s, a)) \cdot 1_{t,h}^{\mathrm{off}}(s, a) \cdot \left(\frac{1}{q_h^{\mathrm{B}}(s, a)} - \frac{1}{\overline{q}_{t,h}^{\mathrm{B}}(s, a) + \frac{\eta}{2} + \sqrt{\frac{\log(2HSAT/\varsigma)}{2t}}}\right)\right]
$$

$$
= \mathbb{E}\left[q_h^{\pi^{\mathrm{C}}, \widetilde{P}_{e(t)}}(s, a)(1 - \ell_{t,h}(s, a))\left(\frac{\overline{q}_{t,h}^{\mathrm{B}}(s, a) + \frac{\eta}{2} + \sqrt{\frac{\log(2HSAT/\varsigma)}{2t}} - q_h^{\mathrm{B}}(s, a)}{\overline{q}_{t,h}^{\mathrm{B}}(s, a) + \frac{\eta}{2} + \sqrt{\frac{\log(2HSAT/\varsigma)}{2t}}}\right)\right]
$$

$$
\leqslant \mathbb{E}\left[q_h^{\pi^{\mathrm{C}}, \widetilde{P}_{e(t)}}(s, a)(1 - \ell_{t,h}(s, a))\frac{\frac{\eta}{2} + 2\sqrt{\frac{\log(2HSAT/\varsigma)}{2t}}}{\overline{q}_{t,h}^{\mathrm{B}}(s, a) + \frac{\eta}{2} + \sqrt{\frac{\log(2HSAT/\varsigma)}{2t}}}\right]
$$

$$
\leqslant \mathbb{E}\left[q_h^{\pi^{\mathrm{C}}, \widetilde{P}_{e(t)}}(s, a)\frac{\frac{\eta}{2} + 2\sqrt{\frac{\log(2HSAT/\varsigma)}{2t}}}{q_h^{\mathrm{B}}(s, a) + \frac{\eta}{2}}\right]
$$

$$
\leqslant \frac{\eta}{2}\mathbb{E}\left[\frac{q_h^{\pi^{\mathrm{C}}, \widetilde{P}_{e(t)}}(s, a)}{q_h^{\mathrm{B}}(s, a)}\right] + 2\sqrt{\frac{\log(2HSAT/\varsigma)}{2t}}\mathbb{E}\left[\frac{q_h^{\pi^{\mathrm{C}}, \widetilde{P}_{e(t)}}(s, a)}{q_h^{\mathrm{B}}(s, a)}\right]
$$

$$
\leqslant \frac{\eta}{2} \cdot \frac{q_h^{\pi^{\mathrm{C}}, P}(s, a)}{q_h^{\mathrm{B}}(s, a)} + 2\sqrt{\frac{\log(2HSAT/\varsigma)}{2t}} \cdot \frac{q_h^{\pi^{\mathrm{C}}, P}(s, a)}{q_h^{\mathrm{B}}(s, a)}, \tag{69}
$$

where the first inequality is because $\overline{q}_{t,h}^{\mathrm{B}}(s, a) - q_h^{\mathrm{B}}(s, a) \leqslant \sqrt{\frac{\log(2HSAT/\varsigma)}{2t}}$ due to the good event; the second inequality is because losses are bounded in $[0, 1]$ and $\overline{q}_{t,h}^{\mathrm{B}}(s, a) + \sqrt{\frac{\log(2HSAT/\varsigma)}{2t}} \geqslant q_h^{\mathrm{B}}(s, a)$; the last inequality is because $q_h^{\pi^{\mathrm{C}}, \widetilde{P}_{e(t)}}(s, a) \leqslant q_h^{\pi^{\mathrm{C}}, P}(s, a)$ (Lemma 3).

Adding all episodes, layers, and state-action pairs together, we get

$$
\mathbb{E}\left[\sum_{t=1}^{T}\langle q^{\pi^{\mathrm{C}}, \widetilde{P}_{e(t)}}, \widehat{\ell}_t - \widetilde{\ell}_t\rangle\right] = \widetilde{O}\left(\eta T \mathcal{C}(\pi^{\mathrm{C}}) + \mathcal{C}(\pi^{\mathrm{C}})\sum_{t=1}^{T}\frac{1}{\sqrt{t}}\right) = \widetilde{O}\left(\eta T \mathcal{C}(\pi^{\mathrm{C}}) + \mathcal{C}(\pi^{\mathrm{C}})\sqrt{T}\right), \tag{70}
$$

which implies that

$$
\mathrm{OMDREG} = \mathbb{E}\left[\sum_{t=1}^{T}\langle q^{\pi_t, \widetilde{P}_{e(t)}} - q^{\pi^{\mathrm{C}}, \widetilde{P}_{e(t)}}, \ell_t - \mathbf{1}\rangle\right] = \widetilde{O}\left(\frac{H^2 SA}{\eta} + \eta T \mathcal{C}(\pi^{\mathrm{C}}) + \mathcal{C}(\pi^{\mathrm{C}})\sqrt{T}\right). \tag{71}
$$

### E.2.2. BOUNDING ERROR$_1$

First of all, since all losses are in $[0, 1]$, we have

$$
\mathrm{ERROR}_1 = \mathbb{E}\left[\sum_{t=1}^{T}\langle q^{\pi^{\mathrm{C}}, \widetilde{P}_{e(t)}} - q^{\pi^{\mathrm{C}}, P}, \ell_t - \mathbf{1}\rangle\right] \leqslant \mathbb{E}\left[\sum_{t=1}^{T}\sum_{h=1}^{H}\sum_{(s,a)\in\mathcal{S}\times\mathcal{A}}\left|q_h^{\pi^{\mathrm{C}}, \widetilde{P}_{e(t)}}(s, a) - q_h^{\pi^{\mathrm{C}}, P}(s, a)\right|\right]. \tag{72}
$$

Our overall strategy is to control this occupancy error using transition estimation error. By Lemma 11, for each epoch $e$ and layer $h = 2, \ldots, H$ we have

$$
\sum_{t=t_e}^{t_{e+1}-1} \sum_{(s,a)\in\mathcal{S}\times\mathcal{A}} \left| q_h^{\pi^{\mathrm{C}},P}(s,a) - q_h^{\pi^{\mathrm{C}},\widetilde{P}_i}(s,a) \right|
$$

$$
\leqslant \sum_{t=t_e}^{t_{e+1}-1} \sum_{k=1}^{h-1} \sum_{(s,a)\in\mathcal{S}\times\mathcal{A}} q_k^{\mathrm{C}}(s,a) \sum_{s'\in\mathcal{S}} \left| P^k(s' \mid s,a) - \widetilde{P}_e^k(s' \mid s,a) \right|
$$

$$
\leqslant \sum_{t=t_e}^{t_{e+1}-1} \sum_{k=1}^{h-1} \sum_{(s,a)\in\mathcal{S}\times\mathcal{A}} q_k^{\mathrm{C}}(s,a) \sum_{s'\in\mathcal{S}} \left( 2\sqrt{\frac{\overline{P}_e^k(s' \mid s,a)}{\max\{1, N_e^k(s,a)\}}} + \frac{14}{3\max\{1, N_e^k(s,a)\}} \right)
$$

$$
\leqslant \sum_{t=t_e}^{t_{e+1}-1} \sum_{k=1}^{h-1} \sum_{(s,a)\in\mathcal{S}\times\mathcal{A}} q_k^{\mathrm{C}}(s,a) \left( 2\sqrt{\sum_{s'\in\mathcal{S}} \overline{P}_e^k(s' \mid s,a)} \sqrt{\frac{S}{\max\{1, N_e^k(s,a)\}}} + \frac{14S}{3\max\{1, N_e^h(s,a)\}} \right)
$$

$$
\leqslant 2\sqrt{S} \sum_{t=t_e}^{t_{e+1}-1} \sum_{k=1}^{h-1} \sum_{(s,a)\in\mathcal{S}\times\mathcal{A}} q_k^{\mathrm{C}}(s,a) \sqrt{\frac{1}{t_e \cdot q_k^{\mathrm{B}}(s,a)}} + \frac{14S}{3} \sum_{t=t_e}^{t_{e+1}-1} \sum_{k=1}^{h-1} \sum_{(s,a)\in\mathcal{S}\times\mathcal{A}} \frac{q_k^{\mathrm{C}}(s,a)}{t_e \cdot q_k^{\mathrm{B}}(s,a)}. \tag{73}
$$

In layer $h = 1$, we have $q_h^{\pi^{\mathrm{C}},P} = q_h^{\pi^{\mathrm{C}},\widetilde{P}_{e(t)}}$ because the learner starts from the same fixed initial state in all episodes, so the occupancy in the first step ($h = 1$) is not relevant to the underlying transition. Taking into account all the layers from 1 up to $H$ and all epochs, we have

$$
\mathrm{ERROR}_1 = \mathbb{E}\left[ \sum_{t=1}^{T} \langle q^{\pi^{\mathrm{C}},\widetilde{P}_{e(t)}} - q^{\pi^{\mathrm{C}},P}, \ell_t - \mathbf{1} \rangle \right] \leqslant 2\sqrt{S}\,\mathbb{E}\left[ \sum_{e=1}^{N_{\mathrm{epoch}}} \frac{1}{\sqrt{t_e}} \sum_{t=t_e}^{t_{e+1}-1} \sum_{h=1}^{H} \sum_{k=1}^{h} \sum_{(s,a)\in\mathcal{S}\times\mathcal{A}} \frac{q_k^{\mathrm{C}}(s,a)}{\sqrt{q_k^{\mathrm{B}}(s,a)}} \right]
$$

$$
+ \frac{14S}{3}\,\mathbb{E}\left[ \sum_{e=1}^{N_{\mathrm{epoch}}} \frac{1}{t_e} \sum_{t=t_e}^{t_{e+1}-1} \sum_{h=1}^{H} \sum_{k=1}^{h} \sum_{(s,a)\in\mathcal{S}\times\mathcal{A}} \frac{q_k^{\mathrm{C}}(s,a)}{q_k^{\mathrm{B}}(s,a)} \right]. \tag{74}
$$

For the first term, we have

$$\mathbb{E}\left[\sum_{e=1}^{N_{\text{epoch}}} \frac{1}{\sqrt{t_e}} \sum_{t=t_e}^{t_{e+1}-1} \sum_{h=1}^{H} \sum_{k=1}^{h} \sum_{(s,a)\in\mathcal{S}\times\mathcal{A}} \frac{q_k^{\text{C}}(s,a)}{\sqrt{q_k^{\text{B}}(s,a)}}\right]$$

$$\overset{(a)}{\leqslant} \mathbb{E}\left[\sum_{e=1}^{N_{\text{epoch}}} \frac{1}{\sqrt{t_e}} \sum_{t=t_e}^{t_{e+1}-1} \sum_{h=1}^{H} \sum_{k=1}^{h} \sqrt{\sum_{(s,a)\in\mathcal{S}\times\mathcal{A}} q_k^{\text{C}}(s,a)} \sqrt{\sum_{(s,a)\in\mathcal{S}\times\mathcal{A}} \frac{q_k^{\text{C}}(s,a)}{q_k^{\text{B}}(s,a)}}\right]$$

$$\overset{(b)}{=} \mathbb{E}\left[\sum_{e=1}^{N_{\text{epoch}}} \frac{1}{\sqrt{t_e}} \sum_{t=t_e}^{t_{e+1}-1} \sum_{h=1}^{H} \sum_{k=1}^{h} \sqrt{\sum_{(s,a)\in\mathcal{S}\times\mathcal{A}} \frac{q_k^{\text{C}}(s,a)}{q_k^{\text{B}}(s,a)}}\right]$$

$$= \mathbb{E}\left[\sum_{e=1}^{N_{\text{epoch}}} \frac{1}{\sqrt{t_e}} \sum_{t=t_e}^{t_{e+1}-1} \sum_{h=1}^{H} (H-h+1)\sqrt{\sum_{(s,a)\in\mathcal{S}\times\mathcal{A}} \frac{q_h^{\text{C}}(s,a)}{q_h^{\text{B}}(s,a)}}\right]$$

$$\overset{(c)}{\leqslant} \mathbb{E}\left[\sum_{e=1}^{N_{\text{epoch}}} \frac{1}{\sqrt{t_e}} \sum_{t=t_e}^{t_{e+1}-1} \sqrt{\sum_{h=1}^{H}(H-h+1)^2}\sqrt{\sum_{h=1}^{H}\sum_{(s,a)\in\mathcal{S}\times\mathcal{A}} \frac{q_h^{\text{C}}(s,a)}{q_h^{\text{B}}(s,a)}}\right]$$

$$\overset{(d)}{\leqslant} \mathbb{E}\left[\sum_{e=1}^{N_{\text{epoch}}} \frac{1}{\sqrt{t_e}} \sum_{t=t_e}^{t_{e+1}-1} H^{3/2}\sqrt{\mathcal{C}(\pi^{\text{C}})}\right]$$

$$\leqslant H^{3/2}\sqrt{\mathcal{C}(\pi^{\text{C}})}\int_{x=1}^{T} x^{-1/2}dx \leqslant H^{3/2}\sqrt{\mathcal{C}(\pi^{\text{C}})T}, \tag{75}$$

where

- we apply Cauchy–Schwarz inequality in step (a) and have

$$\sum_{(s,a)\in\mathcal{S}\times\mathcal{A}} \frac{q_h^{\text{C}}(s,a)}{\sqrt{q_h^{\text{B}}(s,a)}} \leqslant \sqrt{\sum_{(s,a)\in\mathcal{S}\times\mathcal{A}}\left(\sqrt{q_h^{\text{C}}(s,a)}\right)^2} \cdot \sqrt{\sum_{(s,a)\in\mathcal{S}\times\mathcal{A}}\left(\sqrt{\frac{q_h^{\text{C}}(s,a)}{q_h^{\text{B}}(s,a)}}\right)^2};$$

- step (b) is simply because the total occupancy in a layer is one, i.e., $\sum_{(s,a)\in\mathcal{S}\times\mathcal{A}} q_h^{\pi,P}(s,a) = 1, \forall\pi\in\Pi, h\in[H]$;

- step (c) is also due to Cauchy–Schwarz inequality;

- in step (d), we upper bound $\sqrt{\sum_{h=1}^{H}(H-h+1)^2}$ by $H^{3/2}$ and replace $\sum_{h=1}^{H}\sum_{(s,a)\in\mathcal{S}\times\mathcal{A}}\frac{q_h^{\text{C}}(s,a)}{q_h^{\text{B}}(s,a)}$ with its definition $\mathcal{C}(\pi^{\text{C}})$.

For the second term, we have

$$\mathbb{E}\left[\sum_{e=1}^{N_{\text{epoch}}} \frac{1}{t_e} \sum_{t=t_e}^{t_{e+1}-1} \sum_{h=1}^{H} \sum_{k=1}^{h} \sum_{(s,a)\in\mathcal{S}\times\mathcal{A}} \frac{q_k^{\text{C}}(s,a)}{q_k^{\text{B}}(s,a)}\right]$$

$$= \mathbb{E}\left[\sum_{e=1}^{N_{\text{epoch}}} \frac{1}{t_e} \sum_{t=t_e}^{t_{e+1}-1} \sum_{h=1}^{H} (H-h+1)\sum_{(s,a)\in\mathcal{S}\times\mathcal{A}} \frac{q_h^{\text{C}}(s,a)}{q_h^{\text{B}}(s,a)}\right]$$

$$\leqslant \mathbb{E}\left[\sum_{e=1}^{N_{\text{epoch}}} \frac{1}{t_e} \sum_{t=t_e}^{t_{e+1}-1} H\mathcal{C}(\pi^{\text{C}})\right] \leqslant H\mathcal{C}(\pi^{\text{C}})\int_{x=1}^{T} x^{-1}dx \leqslant H\mathcal{C}(\pi^{\text{C}})\log T. \tag{76}$$

Combining two parts together, we have

$$\text{ERROR}_1 = \mathbb{E}\left[\sum_{t=1}^{T}\langle q^{\pi^{\text{C}},\widetilde{P}_{e(t)}} - q^{\pi^{\text{C}},P}, \ell_t - \mathbf{1}\rangle\right] \leqslant 2H^{3/2}\sqrt{S\mathcal{C}(\pi^{\text{C}})T} + \frac{14}{3}SH\mathcal{C}(\pi^{\text{C}})\log T. \tag{77}$$

### E.2.3. BOUNDING ERROR$_2$

By Lemma 4, it holds that

$$\text{ERROR}_2 = \mathbb{E}\left[\sum_{t=1}^{T}\langle q^{\pi_t,P} - q^{\pi_t,\widetilde{P}_{e(t)}}, \ell_t - \mathbf{1}\rangle\right] \leqslant 0. \tag{78}$$

### E.2.4. COMBINING ALL PIECES TOGETHER

Combining all three terms, for any fixed learning rate $\eta > 0$, Algorithm 4 ensures

$$\mathbb{E}\left[\sum_{t=1}^{T}\langle q^{\pi_t,P} - q^{\pi^{\text{C}},P}, \ell_t\rangle\right] = \widetilde{O}\left(\frac{H^2SA}{\eta} + \eta\mathcal{C}(\pi^{\text{C}})T + \mathcal{C}(\pi^{\text{C}})\sqrt{T} + H^{3/2}\sqrt{S\mathcal{C}(\pi^{\text{C}})T}\right), \forall\pi^{\text{C}} \in \Pi. \tag{79}$$

Note that when $T$ is large enough, $(\mathcal{C}(\pi^{\text{C}})\sqrt{T})$ dominates the $(SH\mathcal{C}(\pi^{\text{C}})\log T)$ term in ERROR$_1$.

Similar to the MAB case, if we create $\lfloor\log(HT) + 1\rfloor$ off-policy instances, the $i$-th one ($i \geqslant 1$) among which has learning rate $\eta^i = \frac{H\sqrt{SA}}{\sqrt{T\cdot\exp(i)}}$, then for any $\pi^{\text{C}} \in \Pi$ whose $\mathcal{C}(\pi^{\text{C}}) = O(HT)$, there must exist some $i \geqslant 1$ such that

$$\text{BASEREG}_i = \mathbb{E}\left[\sum_{t=1}^{T}\langle q^{\pi_t^i,P} - q^{\pi^{\text{C}},P}, \ell_t\rangle\right] = \widetilde{O}\left(H\sqrt{\mathcal{C}(\pi^{\text{C}})SAT} + \mathcal{C}(\pi^{\text{C}})\sqrt{T} + H^{3/2}\sqrt{S\mathcal{C}(\pi^{\text{C}})T}\right), \tag{80}$$

where $\{\pi_t^i\}_{t=1,\dots,T}$ is the policy sequence produced by the $i$-th instance ($i \geqslant 1$).

### E.3. Meta-Level Regret

By the same reasoning as in the known transition case, we have

$$\text{METAREG}_i = \mathbb{E}\left[\sum_{t=1}^{T}\langle p_t - u_i, \widehat{c}_t\rangle\right] = \widetilde{O}\left(\frac{1}{\eta^{\text{meta}}} + \eta^{\text{meta}}H^2T + H\sqrt{T} - \frac{\mathbb{E}[\rho_{T,i}]}{\eta^{\text{meta}}}\right), \forall i \geqslant 0. \tag{81}$$

### E.4. Combining Meta and Base Levels together

For any fixed learning rate $\eta^{\text{meta}} > 0$ in Algorithm 8, the following bounds hold universally against all $\pi^{\text{C}} \in \Pi$:

$$\mathbb{E}[R(\pi^{\text{C}})] = \widetilde{O}\left(\mathbb{E}\left[\frac{1}{\eta^{\text{meta}}} + \eta^{\text{meta}}H^2T - \frac{\rho_{T,0}}{\eta^{\text{meta}}} + H\sqrt{T} + (\rho_{T,0})^{1/2}H^2S\sqrt{AT}\right]\right), \tag{82}$$

$$\mathbb{E}[R(\pi^{\text{C}})] = \widetilde{O}\left(\frac{1}{\eta^{\text{meta}}} + \eta^{\text{meta}}H^2T + H\sqrt{T} + \eta^{\text{meta}}H^4S^2AT\right), \tag{83}$$

$$\mathbb{E}[R(\pi^{\text{C}})] = \widetilde{O}\left(\frac{1}{\eta^{\text{meta}}} + \eta^{\text{meta}}H^2T + H\sqrt{T} + H\sqrt{\mathcal{C}(\pi^{\text{C}})SAT} + \mathcal{C}(\pi^{\text{C}})\sqrt{T} + H^{3/2}\sqrt{\mathcal{C}(\pi^{\text{C}})ST}\right). \tag{84}$$

Choosing $\eta^{\text{meta}} = \frac{1}{H\sqrt{T}}$ (note that $\mathcal{C}(\pi^{\text{C}}) \geqslant H$), we get

$$\mathbb{E}[R(\pi^{\text{C}})] = \widetilde{O}\left(\min\{H^3S^2A, H\sqrt{\mathcal{C}(\pi^{\text{C}})SA} + \mathcal{C}(\pi^{\text{C}}) + H^{3/2}\sqrt{S\mathcal{C}(\pi^{\text{C}})}\}\sqrt{T}\right), \forall\pi^{\text{C}} \in \Pi. \tag{85}$$

# F. Numerical Results

We conduct simulations in the MAB setting. We construct a non-stochastic environment following Section 8 of Zimmert & Seldin (2021). The configuration parameters include the time horizon $T$, the number of actions $A$, and the loss gap $\Delta > 0$, where in every round the loss of the optimal action $a^*$ is lower than that of every other action by $\Delta$.

For the behavior policy $\pi^{\mathrm{B}}$, we consider three regimes for the value of $\pi^{\mathrm{B}}(a^*)$, with the remaining probability mass distributed uniformly over the other actions: (i) $\pi^{\mathrm{B}}(a^*) = 0.5$, so that $\mathcal{C}(\pi^*) = 2$; (ii) $\pi^{\mathrm{B}}(a^*) = 1/A$, so that $\mathcal{C}(\pi^*) = A$; and (iii) $\pi^{\mathrm{B}}(a^*) = 1/T$, so that $\mathcal{C}(\pi^*) = T$.

Figure 1 reports the regret trajectory, i.e., the cumulative regret against $\pi^*$ as a function of time/round, under $T = 10^5$, $A = 8$, and $\Delta = 0.0125$, with curves showing the mean and shaded regions showing one standard deviation over 20 independent trials. Other configurations among $A \in \{8, 16, 32\}$ and $\Delta \in \{0.0125, 0.025, 0.05, 0.1\}$ overall exhibit similar phenomena.

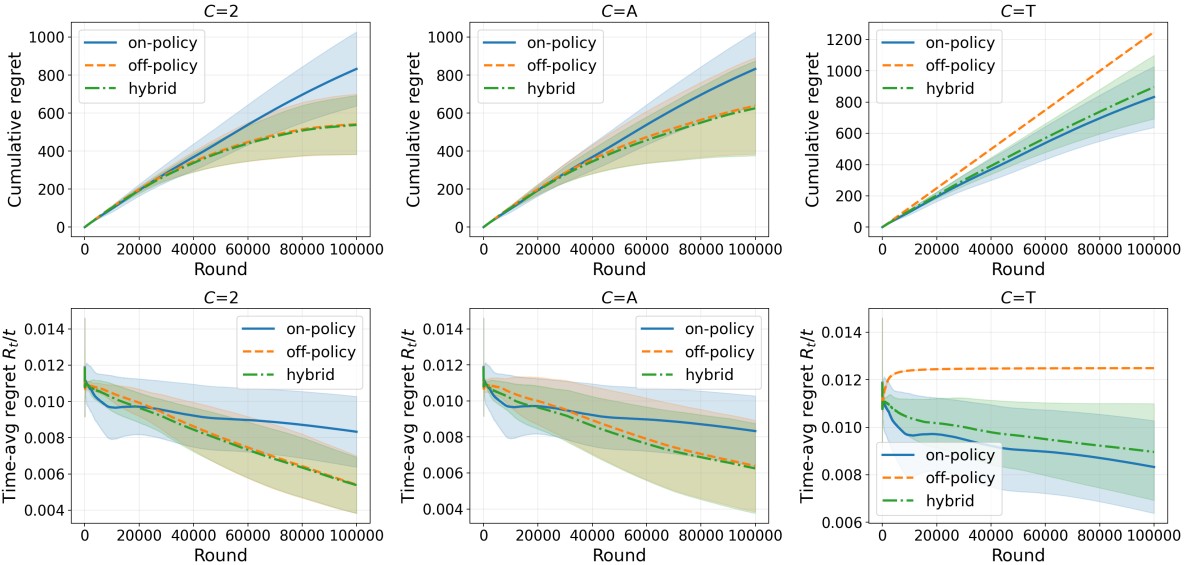

*Figure 1.* Regret trajectory under $T = 10^5$, $A = 8$, and $\Delta = 0.0125$. Each curve shows the mean cumulative regret over time, and shaded regions show one standard deviation over 20 independent trials.

These results support the intended behavior of our hybrid algorithm. When coverage is good, i.e., $\mathcal{C}(\pi^*) = 2$ or $A$, the hybrid method matches or slightly improves upon the off-policy algorithm. This does not contradict our theoretical bounds: the off-policy algorithm uses learning rate $\eta^{\mathrm{IX}} = 1/\sqrt{T}$ and has regret bound $\mathcal{C}(\pi^{\mathrm{C}})\sqrt{T}$, whereas the hybrid algorithm improves this dependence to $\sqrt{\mathcal{C}(\pi^{\mathrm{C}})T}$ in good-coverage regimes. In addition, the hybrid algorithm clearly outperforms the on-policy algorithm. When coverage is very poor, i.e., $\mathcal{C}(\pi^*) = T$, the pure off-policy method deteriorates sharply, whereas the hybrid algorithm remains closer to the on-policy algorithm.

# G. Supporting Lemmas

**Lemma 1** (Regret guarantee of OMD update for MAB). *In the MAB case, for any non-negative loss estimate $\widehat{\ell}_t$ and fixed learning rate $\eta \equiv \eta_t$, the update rule*

$$\pi_{t+1} = \underset{\pi \in \Delta(\mathcal{A})}{\mathrm{argmin}} \left( \langle \pi, \widehat{\ell}_t \rangle + D_{\psi_t^{\mathrm{SE}}}(\pi, \pi_t) \right) \tag{86}$$

*with uniform $\pi_1$ ensures that*

$$\sum_{t=1}^{T} \langle \pi_t - \pi, \widehat{\ell}_t \rangle \leqslant \frac{\log A}{\eta} + \frac{1}{\eta} \sum_{t=1}^{T} \sum_{a \in \mathcal{A}} \pi_t(a) \left( \eta \widehat{\ell}_t(a) - 1 + \exp(-\eta \widehat{\ell}_t(a)) \right), \forall \pi \in \Delta(\mathcal{A}) \tag{87}$$

*for arbitrary loss estimate sequence $\{\widehat{\ell}_t\}_{t=1,\dots,T}$.*

*If all loss estimates are non-negative, then it implies the well-known result*

$$\sum_{t=1}^{T} \langle \pi_t - \pi, \widehat{\ell}_t \rangle \leqslant \frac{\log A}{\eta} + \eta \sum_{t=1}^{T} \sum_{a \in \mathcal{A}} \pi_t(a)(\widehat{\ell}_t(a))^2, \forall \pi \in \Delta(\mathcal{A}). \tag{88}$$

**Lemma 2** (Regret guarantee of OMD update for MDP; Lemma 13 of Jin et al. (2020a))**.** *In the MDP case, suppose that $\mathcal{P}_e$ is a set of transition functions maintained for the $e$-th epoch. The update rule*

$$\widehat{q}_{t+1} = \underset{q \in \mathrm{OM}(\mathcal{P}_{e(t)})}{\operatorname{argmin}} \left( \langle q, \widehat{\ell}_t \rangle + D_{\psi_t^{\mathrm{SE}}}(q, \widehat{q}_t) \right) \tag{89}$$

*with $D_{\psi_t^{\mathrm{SE}}}(x, x') := \frac{1}{\eta} \sum_i \left( x(i) \log\left(\frac{x(i)}{x'(i)}\right) - x(i) + x'(i) \right)$ and uniform $\widehat{q}_1$ ensures against any $q' \in \cap_e \mathrm{OM}(\mathcal{P}_e)$ that*

$$\sum_{t=1}^{T} \langle \widehat{q}_t - q', \widehat{\ell}_t \rangle \leqslant \frac{H \log(H^2 S^2 A)}{\eta} + \frac{1}{\eta} \sum_{t=1}^{T} \sum_{h=1}^{H} \sum_{(s,a) \in \mathcal{S} \times \mathcal{A}} \widehat{q}_t(s,a) \left( \eta \widehat{\ell}_{t,h}(s,a) - 1 + \exp\left(-\eta \widehat{\ell}_{t,h}(s,a)\right) \right) \tag{90}$$

*for arbitrary loss estimate sequence $\{\widehat{\ell}_t\}_{t=1,\dots,T}$.*

*If all loss estimates are non-negative, then it implies the well-known result*

$$\sum_{t=1}^{T} \langle \widehat{q}_t - q', \widehat{\ell}_t \rangle \leqslant \frac{H \log(H^2 S^2 A)}{\eta} + \eta \sum_{t=1}^{T} \sum_{h=1}^{H} \sum_{(s,a) \in \mathcal{S} \times \mathcal{A}} \widehat{q}_t(s,a)(\widehat{\ell}_{t,h}(s,a))^2. \tag{91}$$

**Lemma 3** (Corollary C.8.2 of Jin et al. (2023))**.** *Consider Algorithm 4. In each epoch $e$, for any transition function $P' \in \mathcal{P}_e$, it holds for any policy $\pi$ and layer $h \in [H]$ that*

$$q_h^{\pi, \widetilde{P}_e}(s,a) \leqslant q_h^{\pi, P'}(s,a), \forall (s,a) \in \mathcal{S} \times \mathcal{A}. \tag{92}$$

**Lemma 4** (*cf.* Lemma C.8.3 of Jin et al. (2023))**.** *Consider Algorithm 4. In each epoch $e$, for any transition function $P' \in \mathcal{P}_e$ and any function $f : [H] \times \mathcal{S} \times \mathcal{A} \to [0, +\infty)$, it holds that*

$$\sum_{h=1}^{H} \sum_{(s,a) \in \mathcal{S} \times \mathcal{A}} \left( q_h^{\pi, \widetilde{P}_i}(s,a) - q_h^{\pi, P'}(s,a) \right) f(h,s,a) \leqslant 0. \tag{93}$$

*If function $f$ outputs non-positive values instead, then we have*

$$\sum_{h=1}^{H} \sum_{(s,a) \in \mathcal{S} \times \mathcal{A}} \left( q_h^{\pi, \widetilde{P}_i}(s,a) - q_h^{\pi, P'}(s,a) \right) f(h,s,a) \geqslant 0. \tag{94}$$

**Lemma 5** (Bernstein-type confidence set for transition function estimation; Lemma 2 of Jin et al. (2020a))**.** *Consider Algorithm 4. With probability at least $1 - \zeta$, we have the true transition function $P \in \cap_e \mathcal{P}_e$, where*

$$\mathcal{P}_e := \left\{ P' : \left| P'^h(s' \mid s,a) - \overline{P}^h(s' \mid s,a) \right| \leqslant \phi_e^h(s,a,s'), \forall h \in [H], (s,a,s') \in \mathcal{S} \times \mathcal{A} \times \mathcal{S} \right\} \tag{95}$$

*with*

$$\phi_e^h(s,a,s') := 2\sqrt{\frac{\overline{P}_e^h(s' \mid s,a) \log(4THSA/\zeta)}{\max\{1, N_e^h(s,a)\}}} + \frac{14 \log(4THSA/\zeta)}{3 \max\{1, N_e^h(s,a)\}}. \tag{96}$$

**Lemma 6** (Bacchiocchi et al. (2024))**.** *Consider Algorithm 4. With probability at least $1 - \zeta$, it holds that*

$$\left| \overline{q}_{t,h}^{\mathrm{B}}(s,a) - q_h^{\mathrm{B}}(s,a) \right| \leqslant \sqrt{\frac{\log(HSAT/\zeta)}{t}}, \forall (t,h,s,a) \in [T] \times [H] \times \mathcal{S} \times \mathcal{A}. \tag{97}$$

**Lemma 7** (*cf.* Appendix A.1 of Gabbianelli et al. (2023))**.** *Let c be any positive constant. For any function $f : \mathcal{S} \times \mathcal{A} \to [0, 1]$ and any valid occupancy q, it holds that*

$$\frac{f(s, a)}{q(s, a) + c} \leqslant \frac{1}{2c} \log \left( 1 + 2c \frac{f(s, a)}{q(s, a)} \right). \tag{98}$$

**Lemma 8** (Upper bound on the number of epochs; Lemma D.3.12 of Jin et al. (2021a))**.** *Algorithm 4 ensures that*

$$N_{\text{epoch}} \leqslant 4HSA(1 + \log T). \tag{99}$$

**Lemma 9** (*cf.* Lemma A.1 of Xie et al. (2021))**.** *With probability at least $1 - \zeta$, for any epoch e, layer h, and state-action pair $(s, a)$, Algorithm 4 ensures that*

$$\frac{q_h^{\text{B}}(s, a)}{\max\{1, N_e^h(s, a)\}} \leqslant \frac{8 \log(HSAT/\zeta)}{t_e}. \tag{100}$$

**Lemma 10** (Meta-level regret; *cf.* Lemma 13 of Agarwal et al. (2017))**.** *In the MAB case, for any base algorithm $i \geqslant 0$, Algorithm 3 ensures that*

$$\mathbb{E}\left[ \sum_{t=1}^{T} \langle p_t - u_i, \widehat{c}_t \rangle \right] = \widetilde{O} \left( \frac{N_{\text{base}}}{\eta^{\text{meta}}} + \eta^{\text{meta}} T - \frac{\mathbb{E}\left[\rho_{T,i}\right]}{\eta^{\text{meta}}} \right). \tag{101}$$

*In the MDP case, for any base algorithm $i \geqslant 0$, Algorithms 7 and 8 ensure that*

$$\mathbb{E}\left[ \sum_{t=1}^{T} \langle p_t - u_i, \widehat{c}_t \rangle \right] = \widetilde{O} \left( \frac{N_{\text{base}}}{\eta^{\text{meta}}} + \eta^{\text{meta}} T H^2 - \frac{\mathbb{E}\left[\rho_{T,i}\right]}{\eta^{\text{meta}}} \right). \tag{102}$$

**Lemma 11** (Lemma B.2 of Rosenberg & Mansour (2019))**.** *Let $\{\pi_t\}_{t=1,\dots,T}$ be a collection of arbitrary policies and $\{P_t\}_{t=1,\dots,T}$ be a collection of arbitrary transition functions. For the true transition P and any layer $h \geqslant 2$, it holds that*

$$\sum_{t=1}^{T} \sum_{(s,a) \in \mathcal{S} \times \mathcal{A}} \left| q_h^{\pi_t, P}(s, a) - q_h^{\pi_t, P_t}(s, a) \right| \leqslant \sum_{t=1}^{T} \sum_{k=1}^{h-1} \sum_{(s,a) \in \mathcal{S} \times \mathcal{A}} q_k^{\pi_t, P}(s, a) \sum_{s' \in \mathcal{S}} \left| P^k(s' \mid s, a) - P_t^k(s' \mid s, a) \right|. \tag{103}$$

