# OpenReview forum: "Hybrid Reinforcement Learning in Adversarial Markov Decision Processes"
_ICML.cc/2026/Conference — ICML 2026 regular_

### Official Review · Reviewer_tSeq · 2026-03-12

**Soundness:** 3
**Presentation:** 3
**Significance:** 3
**Originality:** 3
**Overall Recommendation:** 4
**Confidence:** 3

**Summary:**

The paper proposes a hybrid approach, i.e. which allows on-policy and off-policy feedback, in an MAB, MDP with known transition functions, and MDP with unknown transitions. The authors study the "adversarial" setting, where losses vary time-dependently and arbitrarily rather than dependent on the policy distribution. The algorithm makes use of Corral, a meta-algorithm that takes different base-learners and adapts to the best one. The approach allows for single-policy concentrability, i.e. to allow to depend on the coverage ratio of a behaviour policy rather than that of the entire function class, and improves on previous regret bounds.

**Compliance With Llm Reviewing Policy:**

Affirmed.

**Final Justification:**

The authors have responded well to most of the concerns, including empirical experiments, clarifications, and related work. I still find the discussion of directly related work (particularly in adversarial MDPs) to be limited and lack recent work, and since the changes to the paper are pending it is with some caution that I accept.

**Key Questions For Authors:**

- There is limited discussion of related work and it becomes difficult to assess the originality. There is only a single alternative regret bound provided for each category and some of those are 10 years old. A broader overview with more recent regret bounds and a discussion of the bounds (in related settings) along with the trade-offs they represent would give more insight into the significance of the work, and the challenges that were solved. The stated comparison in table 1 is pretty limited, although this may be because the "non-stochastic" setting is rare. It would be good though to discuss the bounds for techniques with singe-policy vs class-level concentrability and techniques designed for stochastic settings.

- The distinction of the non-stochastic/adversarial setting vs the stochastic setting could be clarified formally in the text.

- Table 1:
   - Why do your results in denote $C(\pi*)$  instead of $C(\pi^C)$ ? It seems to hide the key difference of single-policy concentrability, or do all the stated $C(\pi^*)$ in the table refer to single-policy concentrability?
   - It is not clear under which assumptions these algorithms work (e.g. are they all in the adversarial setting)

- The paper uses the expectation over the regret throughout the paper. It should be clarified what the expectation is taken over and what the regret would look like when removing this expectation.

- Maybe good to clarify “this attempt” at the end of section 4.3.1

- L.8 and 11 of algorithm 4 seem to repeat the same operation with one additional argument. What is the point? Is this to have separate operators for two and three argument functions, and the two-argument function would average across the third dimension?

- The paper does not present any experiments. It may be useful to provide some illustrative experiments or comment on which applications would be particularly suited for the proposed learning scheme.

- The technique makes log(T) copies of off-policy algorithm. Could the authors provide some intuitive reasoning about why this is needed? Why does at least one instance provide the base-level regret $\tilde{O}(\sqrt{C(\pi^C)T})$?


- Some attention to grammatical correctness could improve the paper ,e.g.
“From occupancy view” --> From the view point of occupancy,
“When true transition P is known” --> when the true transition dynamics model P is known
lemma 11: polices --> policies
transitions -> transition functions
(note that there are many different examples)

**Limitations:**

Yes

**Strengths And Weaknesses:**

Strengths:
- The paper makes clear formalisations and derivations, and is adequately structured.
- The stated results depend on the square root of the coverage and provide single-policy concentrability.

Weaknesses:
- There is no practical demonstration.
- The discussion of the related work is quite limited.
- The paper is dense and could benefit from a few clarifying explanations as well as a few grammatical corrections.

---

> ### Author Rebuttal · Authors · 2026-03-30
>
> Thanks for comments and questions. Below, we address them each point in detail.
>
> > **Comment 1**: It would be good though to discuss techniques for stochastic settings.
>
> **Re**: Thank you for this helpful suggestion. We will expand the related-work discussion and provide more technical detail on the literature most relevant to offline/off-policy RL in stochastic settings.
>
> > **Comment 2**: Why does Table 1 denote $C(\pi^*)$ instead of $C(\pi^C)$?
>
> **Re**: Thank you for raising this point. All bounds in Table 1 are indeed derived for an arbitrary $\pi^C$, i.e., they are **universal** and enjoy single-policy concentrability, which means that letting $\pi^C = \pi^*$ as in Table 1 is just a special case of the results we derive.
>
> Our rationale was to use a single regret definition uniformly across all setups and feedback models in the table, even though in the on-policy setting one often focuses only on regret with respect to the strongest comparator, namely $\pi^*$. That said, we agree with the reviewer that this presentation may obscure the role of **single-policy concentrability**, and we will revise the table and surrounding discussion in the next version to make this distinction more explicit.
>
> > **Comment 3**: It should be clarified what the expectation is taken over and what the regret would look like when removing this expectation.
>
> **Re**: Throughout the paper, the expectation in the regret is taken over the randomness of the learning algorithm and the trajectory generation induced by the interaction process (e.g., action sampling and state transitions). Correspondingly, if we remove the expectation, the notion becomes the **path-wise / realized regret**, namely the regret defined for a single realization of the algorithm and environment randomness.
>
> Since our learning algorithm is randomized, it is generally not meaningful to focus on a single sample path. Instead, the standard notions are **expected regret** and **high-probability regret**, with the latter providing a stronger guarantee.
>
> > **Comment 4**: L.8 and 11 of algorithm 4 seem to repeat the same operation with one additional argument.
>
> **Re**: Thank you for pointing this out. In implementation, it suffices to maintain the three-argument count $N_e^h(s,a,s')$, since the two-argument count is simply its marginal: $N_e^h(s,a)=\sum_{s'\in\mathcal{S}} N_e^h(s,a,s')$.
>
> Both quantities are used in defining the confidence set for the true transition kernel $P$; please see Eqs. (18)-(20). We agree that this relationship should be presented more clearly, and we will improve the exposition in the next version.
>
> > **Comment 5**: Intuition about $\log(T)$ copies of off-policy algorithm.
>
> **Re**: In short, the reason for using multiple copies is to obtain the sharp $\sqrt{C(\pi^C)}$ dependence in the hybrid regret bound.
>
> Indeed, our Theorem 1 and Remark 1 show that, for a single off-policy base algorithm, obtaining regret $O(\sqrt{C(\pi^C)T})$ against a comparator $\pi^C$ requires the learning rate $\eta^{IX}=\Theta\left(\frac{1}{\sqrt{C(\pi^C)T}}\right)$.
>
> The challenge is to tune $\eta^{IX}$ with unknown $C(\pi^C)$. For example, even for the strongest comparator $\pi^\*$, directly setting $\eta^{IX}=1/\sqrt{C(\pi^*)T}$ is **infeasible** because it would require prior knowledge of optimal action $a^\*$. Without this knowledge, one would only use the generic choice $\eta^{IX}=1/\sqrt{T}$, which yields the weaker bound $O(C(\pi^\*)\sqrt{T})$.
>
> Our solution is therefore to construct an exponential grid over $[1,T]$. This is sufficient because the relevant coverage ratio lies in this interval, so covering $[1,T]$ guarantees that it is captured up to a constant factor.
>
> By this doubling trick-type design, it requires only $O(\log T)$ grid points, and hence only $O(\log T)$ instances of the off-policy algorithm. The tradeoff is that maintaining these $\log T$ copies introduces additional $\log T$ factors in the meta-level regret.
>
> Assigning one learning rate to each grid point ensures that, for any comparator $\pi^C$, there exists at least one copy whose tuning satisfies $\eta^{IX}=\Theta(\frac{1}{\sqrt{C(\pi^C)T}})$ up to constants, which in turn yields the base-level regret $O(\sqrt{C(\pi^C)T})$. See the proof of Theorem 3, especially Eqs. (31) and (32).
>
> Corral is then used at the meta level to adapt online to this well-tuned copy, which allows the final hybrid regret to retain the desired $O(\sqrt{C(\pi^C)T})$ (universally for all $\pi^C$'s which include the unknown $\pi^*$; when $C(\pi^C)$ is sufficiently small).
>
> > **Comment 6**: Experiments
>
> **Re**: We've conducted simulations in the MAB setting to illustrate the effectiveness of our hybrid learning algorithm. Please see **Comment 4** in our response to **Reviewer PL5R**
>
> > **Comment 7**: assumptions
>
> **Re**: These results hold when the losses are generated in an **oblivious** manner, i.e., $\\{\ell_t\\}_{t=1}^T$ is fixed before the interaction begins, as stated in Lines 97 and 307.

---

> > ### Author Rebuttal · Reviewer_tSeq · 2026-04-02
> >
> > The authors have responded to all of my questions and added illustrative experiments, showing the technique can be effective in practice.
> >
> > The only remaining concern is the related work update is not yet visible so it cannot yet be assessed. I do not know the relevant literature in sufficient depth to know of any omissions so I am not sure whether there are any significant omissions.

---

> > > ### Author Response · Authors · 2026-04-02
> > >
> > > Dear Reviewer tSeq,
> > >
> > > Thank you for your follow-up question, which gives us an opportunity to further clarify and emphasize our contributions.
> > >
> > > # Summary
> > >
> > > In short, achieving proper pessimism and establishing single-policy concentrability for unknown-transition MDPs in the adversarial setting involves a **fundamental challenge**, because the stochastic and adversarial settings rely on **different algorithmic frameworks**. Below, we provide additional technical clarification.
> > >
> > > # Stochastic setting
> > >
> > > ## All-policy concentrability
> > > In the **stochastic** setting, there exist offline RL algorithms that either do not incorporate **pessimism** at all (i.e., they act greedily with respect to empirical estimates) or do not incorporate it strongly enough. As a result, their theoretical guarantees can be suboptimal and/or require stronger coverage assumptions, which are less likely to hold in practice; see, for example, [6, 7].
> > >
> > > ## Single-policy concentrability
> > > To this end, one plausible way to implement the pessimism principle is to **subtract a penalty term**, rather than add an exploration bonus, to the empirical estimates. This penalizes actions (or state-action pairs) with greater uncertainty more heavily, rather than encouraging them. Put differently, this can be viewed as flipping the sign of the bonus term in the canonical UCB (upper confidence bound) framework, thereby effectively constructing an LCB (lower confidence bound) index [1].
> > >
> > > With a suitably designed Bernstein-type negative bonus term, the PEVI-Adv algorithm proposed in [2] achieves the minimax-optimal sample complexity for episodic tabular MDPs, improving upon the result in [4], which uses a Hoeffding-type bonus. More broadly, [3] introduced a refined notion of the (single-policy) concentrability coefficient for both finite- and infinite-horizon tabular MDPs, still within the LCB-based framework.
> > >
> > > # Adversarial setting
> > > In contrast, in the **non-stochastic/adversarial** setting, the algorithmic framework shifts from UCB/LCB-style methods to FTRL/OMD. As a result, it becomes highly nontrivial to implement pessimism in the same manner as in the stochastic setting. More specifically, in the adversarial setting, there is **no explicit UCB-type bonus term in the OMD objective that can simply be sign-flipped** to induce sufficient pessimism for handling transition-estimation uncertainty.
> > >
> > > This was indeed one of the main technical challenges we encountered in this work. To address it, our approach is to inject pessimism directly into the **occupancy measure space**, over which the OMD update is performed, through a **biased transition function** [5]. Although this technique was originally proposed with an optimistic flavor to encourage exploration in the on-policy setting, from the reward perspective it becomes pessimistic, and thus is particularly well suited to our purpose here. It would be interesting to explore whether there are alternative ways to introduce pessimism for unknown transitions, for example by modifying the OMD objective directly.
> > >
> > > **References**
> > >
> > > [1] Jin, Ying, Zhuoran Yang, and Zhaoran Wang. "Is pessimism provably efficient for offline RL?." ICML 21. https://arxiv.org/abs/2012.15085
> > >
> > > [2] Xie, Tengyang, et al. "Policy finetuning: Bridging sample-efficient offline and online reinforcement learning." NeurIPS 21. https://arxiv.org/abs/2106.04895
> > >
> > > [3] Li, Gen, et al. "Settling the sample complexity of model-based offline reinforcement learning." The Annals of Statistics. https://arxiv.org/abs/2204.05275
> > >
> > > [4] Rashidinejad, Paria, et al. "Bridging offline reinforcement learning and imitation learning: A tale of pessimism." NeurIPS 21. https://arxiv.org/abs/2103.12021
> > >
> > > [5] Jin, Tiancheng, et al. "No-regret online reinforcement learning with adversarial losses and transitions." NeurIPS 23. https://arxiv.org/abs/2305.17380
> > >
> > > [6] Liu, Yao, et al. "Provably good batch off-policy reinforcement learning without great exploration." NeurIPS 20.https://arxiv.org/abs/2007.08202
> > >
> > > [7] Riedmiller, Martin. "Neural fitted Q iteration–first experiences with a data efficient neural reinforcement learning method." ECML 05. https://link.springer.com/chapter/10.1007/11564096_32

---

### Official Review · Reviewer_dQzv · 2026-03-13

**Soundness:** 4
**Presentation:** 3
**Significance:** 3
**Originality:** 3
**Overall Recommendation:** 5
**Confidence:** 4

**Summary:**

This paper studies hybrid RL simultaneously leveraging on-policy and off-policy feedback in adversarial (non-stationary) MDPs with unknown transitions. The core contributions are: A new hybrid RL framework based on the CORRAL meta-algorithm that adaptively combines on-policy and off-policy base learners under adversarial losses, preserving worst-case guarantees while exploiting coverage-dependent improvements when coverage is favorable. The hybrid regret bound (Theorem 6) smoothly interpolates between the on-policy worst-case and the off-policy coverage-dependent guarantee. The key technical novelty is adapting the biased transition function to introduce pessimism rather than optimism, making the error term non-positive and bounding all remaining terms through C(pi*).

**Compliance With Llm Reviewing Policy:**

Affirmed.

**Final Justification:**

The rebuttal addressed most of my concerns. I keep positive about this paper.

**Key Questions For Authors:**

- Is the sqrt(A) overhead in the hybrid MAB bound tight?

- Why does Theorem 5 carry an SA factor in the leading term relative to the known-transition Theorem 9? Is this a fundamental barrier imposed by unknown transitions, or is it an artifact of the epoch-doubling analysis? Specifically: could the H^(3/2)*sqrt(S*C(\pi*)*T) term from ERROR1 be the dominant term in regimes of practical interest, and if so, is that term improvable?

- In Algorithm 4 (unknown transition, off-policy), q^B_h(s,a) is unknown and replaced by an online estimate q^B_{t,h}(s,a). The confidence interval in Lemma 6 requires the behavior policy to generate sufficiently many visits. Are there regimes (sparse behavior policy, small q^B values) where the online estimation error dominates, and how does this affect the C(pi*) dependence?

- All results are in the tabular MDP setting. Are there fundamental obstacles to extending the biased transition pessimism technique to linear or general function approximation? In particular, does the epoch-based doubling argument survive when the occupancy space is no longer finite-dimensional?

**Limitations:**

yes

**Strengths And Weaknesses:**

Strengths:

1. Single-policy concentrability under unknown transitions in adversarial MDPs is not well studied yet. The affirmative resolution is clean and well-motivated. The proof strategy, repurposing the biased transition function to achieve pessimism rather than optimism, is interesting.
2. Theorem 6 achieves both worst-case robustness (O-tilde(H^3 S^2 A * sqrt(T)) regardless of C(pi*)) and coverage-dependent improvement (O-tilde(H*sqrt(SAC(pi*)*T)) when coverage is good). The two guarantees are unified in a single algorithm without prior knowledge of C(pi*), which is the key difficulty addressed by the CORRAL-based design.
3. The MAB section (Section 3) is well-structured and serves its pedagogical purpose. Theorem 4 establishes the min{A, sqrt(C(pi*))} dependence clearly, and the discussion of why stochastic hybrid RL designs break under adversarial losses (lack of stationarity, failure of confidence-bound and stage-based methods) is concise and accurate.

Weaknesses:
W1. The paper establishes that its hybrid bound is simultaneously at least as good as on-policy (worst-case) and at most as bad as off-policy (coverage-dependent). However, there is no lower bound showing that the extra sqrt(A) factor in the MAB hybrid result (Theorem 4, O-tilde(A*sqrt(T)) vs on-policy O-tilde(sqrt(AT))) is unavoidable in the hybrid setting. Without this, it is unclear whether the cost of corralling is fundamental or an artifact of the CORRAL analysis.

W2. Theorem 5 (off-policy, unknown transition) yields dominant term O-tilde(H*sqrt(SAC(pi*)*T)), whereas the known-transition analogue (Theorem 9, Bacchiocchi et al.) achieves O-tilde(sqrt(HC(pi*)*T)) without the SA factor. The paper does not discuss whether this gap is fundamental to unknown transitions or an artifact of the epoch-based analysis.

W3. The paper uses a large number of symbols that are reused across MAB and MDP sections with subtle changes in meaning (e.g., C(pi) defined differently in Eq. 2 vs Eq. 13; q^B as a fixed quantity in the known-transition case but requiring online estimation in Algorithm 4). A consolidated notation table in the appendix would significantly improve readability for non-specialist readers.

W4. This is a theoretical paper and experiments are not standard at this level of the theory track. However, even a single illustrative simulation demonstrating the interpolation between on-policy and off-policy regimes as a function of C(pi*) would strengthen the empirical accessibility of the results, particularly for readers less familiar with regret bounds.

---

> ### Author Rebuttal · Authors · 2026-03-30
>
> Thanks for comments and questions. Below, we address them each point in detail.
>
> > **Comment 1**: There is no lower bound showing that the extra $\sqrt{A}$ factor in the MAB hybrid result (Theorem 4) is unavoidable. Without this, it is unclear whether the cost of corralling is fundamental or an artifact of the CORRAL analysis.
>
> **Re**: Thank you for raising this important question. We agree that this is an important open problem emerging from our work. In fact, we explicitly discuss it in Appendix B, although we were unable to include it in the main text due to the page limit.
>
> Resolving this question would be highly informative for a fundamental understanding of the role of hybrid feedback in the non-stochastic setting. In particular, an ideal next step would be to determine whether the current bound can be tightened, either by designing an algorithm with an improved upper bound of $O(\sqrt{T \cdot \min\\{C(\pi^C), A\\} })$, or by establishing a matching lower bound of $\Omega (\sqrt{T}\cdot \min\\{\sqrt{C(\pi^C)}, A\\} )$.
>
> We will revise the paper to make this point clearer.
>
> > **Comment 2**: The off-policy bound gap between known and unknown transition.
>
> **Re**: We thank the reviewer for this important question. At a high level, when the transition is unknown, the regret analysis typically decomposes into two parts: (i) an online learning regret term (arising from the FTRL/OMD analysis), and (ii) a transition-estimation error term.
>
> For part (i), the OMD regret we derive is worse than that in in the known-transition case, which is because the epoch-based approach we adopt (which seems necessary to apply the biased transition function trick for single-policy concentrability).
>
> For part (ii), it is essentially zero in the known transition case, while we bound it by $O(H^{3/2}\sqrt{ST C(\pi^C)})$ with unknown transition. It wouldn't be dominating, as that would require $C(\pi^C) = O(H/A)$ through calculations, which is impossible as we have $C(\pi^C)  = \Omega(H)$
>
> Whether this worse bound is unavoidable, or can be improved by a sharper analysis or a different algorithm, remains an open question. In particular, establishing a matching lower bound for the unknown-transition setting is an important future direction. We will clarify this point in the revision.
>
> > **Comment 3**: The paper uses a large number of symbols that are reused across MAB and MDP sections with subtle changes in meaning. A consolidated notation table in the appendix would significantly improve readability for non-specialist readers.
>
> **Re**: Thank you for this helpful suggestion. We will add a notation table in the appendix to make the presentation easier to follow. We believe this will help readers quickly locate the meaning of each symbol.
>
> > **Comment 4**: Experiments
>
> **Re**: We've conducted simulations in the MAB setting to illustrate the effectiveness of our hybrid learning algorithm. Please see **Comment 4** in our response to **Reviewer PL5R**.
>
> > **Comment 5**: In Alg. 4 (unknown transition, off-policy), $q^B$ is unknown and replaced by an online estimate $q^B_{t,h}$. The confidence interval in Lemma 6 requires the behavior policy to generate sufficiently many visits. Are there regimes (sparse behavior policy, small $q^B$ values) where the online estimation error dominates, and how does this affect the C(pi*) dependence?
>
> **Re**: To answer your question directly, the additional regret caused by the online estimation procedure is always bounded by $O(C(\pi^C)\sqrt{T})$, **for any** $q^B$ (or $\pi^B$). Therefore, even when $q^B_h(s,a)$ is arbitrarily small for some $(h,s,a)$ triple, the corresponding regret does **not** become unbounded.
>
> The key reason is that, as established in Lemma 6, the estimation error at episode $t$ decays at the rate $\frac{1}{\sqrt{t}}$, rather than depending on total visitation count of a specific $(h,s,a)$ triple. Summing this error over time, as done in our Eqs. (68) and (69), leads to the overall regret contribution $O(C(\pi^C)\sqrt{T})$.
>
> > **Comment 6**: Extension beyond tabular settings.
>
> **Re**: Please see the **Comment 1** in our response to **Reviewer 3wnQ** due to character limit.
>
> **References**
>
> [1] Jin, Chi, et al. "Learning adversarial markov decision processes with bandit feedback and unknown transition." ICML 20. https://arxiv.org/abs/1912.01192

---

> > ### Author Rebuttal · Reviewer_dQzv · 2026-04-03
> >
> > I thank the authors for the response. Most of my questions are resolved, and I keep my positive score.

---

> > > ### Author Response · Authors · 2026-04-03
> > >
> > > Dear Reviewer dQzv,
> > >
> > > Thank you for your encouraging rating. We are glad to hear that our response has addressed your comments. We will take your feedback into account in future revisions.

---

### Official Review · Reviewer_3wnQ · 2026-03-13

**Soundness:** 3
**Presentation:** 4
**Significance:** 2
**Originality:** 3
**Overall Recommendation:** 5
**Confidence:** 3

**Summary:**

This paper studies the setting of Hybrid RL making use of offline feedback from a fixed behavior policy and online feedback from the currently executed policy in Adversarial MDPs with loss functions changing over time and possibly unknown transition functions. They begin with an Adversarial Multi-Armed Bandit setting. They show how to solve settings including unknown transition functions for Adversarial MDPs and they provide algorithms which achieve single policy concentrability in such settings.

**Compliance With Llm Reviewing Policy:**

Affirmed.

**Final Justification:**

The reviewers added experiments based on the reviewer requests despite being a primarily theoretical paper and explained thoughtfully how they might try to extend to settings with adaptive adversaries. They also motivated their settings well and achieved single policy concentrability with unknown transitions, which is interesting.

**Key Questions For Authors:**

1. How might one extend these results beyond the tabular setting?
2. What about to settings with an adaptive adversary?
3. What are experimental settings that would motivate this?

**Limitations:**

yes

**Strengths And Weaknesses:**

Strengths
- Good presentation of contributions and significance of the setting/regret bounds and discussion of other attempts that failed
- Interesting adversarial loss settings
- Broad theoretical approach seems sound
- Single policy concentrability regret bounds as opposed to all policy concentrability

Weaknesses
- Primarily seems to be studied for tabular MDPs or MABs with finite action spaces
- No experiments provided (I recognize this paper's contribution is of a primarily theoretical nature)
- Primarily seems to be studied for oblivious adversaries as opposed to adaptive--would be helpful to motivate examples of such settings

---

> ### Author Rebuttal · Authors · 2026-03-30
>
> We thank the reviewer's comments and suggestions. Below, we respond to each point in detail.
>
> > **Comment 1**: How might one extend these results beyond the tabular setting?
>
> **Re**: This is indeed a challenging and interesting direction for future work. For the on-policy component, a number of results are already available even in fairly general sequential decision-making settings [1-4].
>
> In contrast, the off-policy case remains much less understood. Even the linear MDP setting appears far from settled, although [5] provides some positive results for linear contextual bandits. More concretely, in the unknown-transition setting, extending our biased transition function to linear MDPs seems highly nontrivial.
>
> Finally, if suitable on-policy and off-policy subroutines can be developed, one may still hope to combine them through a Corral-type meta-algorithm to obtain a corresponding hybrid regret guarantee.
>
> > **Comment 2**: What about to settings with an adaptive adversary?
>
> **Re**: Generally speaking, under bandit feedback, a standard route to obtaining regret guarantees against an adaptive adversary is to: (i) first establish a high-probability bound against any fixed comparator action/policy under an oblivious adversary; and then (ii) take a union bound over all comparator actions/policies. We refer the reviewer to [6, 7] for representative references.
>
> Following this approach, the first key step in our setting would be to derive a high-probability regret bound for the off-policy component. This is already explicitly highlighted as an open problem in Section 5 of [5].
>
> Since high-probability guarantees for the on-policy component are relatively well understood [6, 7], the remaining challenge would be to ensure that the meta-level regret of the hybrid aggregation scheme also holds with high probability. Resolving this would be another essential ingredient toward handling adaptive adversaries.
>
> > **Comment 3**: What are experimental settings that would motivate the oblivious setting?
>
> **Re**: To answer this question directly, one important class of practical scenarios consists of environments that evolve over time for **external** reasons, but **not strategically** in response to the learner's actions. Examples include traffic patterns, wireless channel conditions, demand fluctuations, and news trends.
>
> Another relevant class is **large-population** settings, in which any **single** learner has only a negligible effect on the overall environment, such as large-scale advertising, recommendation systems, or networked systems.
>
> More broadly, we view the **oblivious** setting as a natural first step beyond the potentially restrictive stationary assumption, since it already allows the loss sequence to be fully nonstationary and need not follow any predictable pattern. We agree that extending the analysis to **adaptive adversaries** is an important future direction, although doing so requires overcoming substantial technical challenges, as discussed above.
>
> > **Comment 4**: Experiments.
>
> **Re**: We've conducted simulations in the MAB setting to illustrate the effectiveness of our hybrid learning algorithm. Please see **Comment 4** in our response to **Reviewer PL5R**.
>
> **References**
>
> [1] Liu, H., Wei, C.-Y., & Zimmert, J. An Improved Model-free Decision-estimation Coefficient with Applications in Adversarial MDPs. ICLR 26. https://arxiv.org/abs/2510.08882
>
> [2] Liu, H., Wei, C. Y., & Julian, Z. Decision Making in Hybrid Environments: A Model Aggregation Approach. COLT 25. https://arxiv.org/abs/2502.05974
>
> [3] Liu, H., Wei, C. Y., & Zimmert, J. Towards Optimal Regret in Adversarial Linear MDPs with Bandit Feedback. ICLR 24. https://arxiv.org/abs/2310.11550
>
> [4] Foster, D. J., Rakhlin, A., Sekhari, A., & Sridharan, K. On the complexity of adversarial decision making. NeurIPS 22. https://arxiv.org/abs/2206.13063
>
> [5] Gabbianelli, G., Neu, G., & Papini, M. Online learning with off-policy feedback. ALT 23. https://proceedings.mlr.press/v201/gabbianelli23a.html
>
> [6] Lee, C. W., Luo, H., Wei, C. Y., & Zhang, M. Bias no more: high-probability data-dependent regret bounds for adversarial bandits and mdps. NeurIPS 20. https://arxiv.org/abs/2006.08040
>
> [7] Zimmert, J., & Lattimore, T. Return of the bias: Almost minimax optimal high probability bounds for adversarial linear bandits. COLT 22. https://proceedings.mlr.press/v178/zimmert22b.html

---

> > ### Author Rebuttal · Reviewer_3wnQ · 2026-04-03
> >
> > Thanks--I updated my score based on these responses!

---

> > > ### Author Response · Authors · 2026-04-03
> > >
> > > Dear Reviewer 3wnQ,
> > >
> > > Thank you for your positive rating. We are pleased that our response addressed your comments. We will carefully incorporate your feedback into future revisions.

---

### Official Review · Reviewer_PL5R · 2026-03-13

**Soundness:** 3
**Presentation:** 3
**Significance:** 3
**Originality:** 3
**Overall Recommendation:** 4
**Confidence:** 2

**Summary:**

The paper studies hybrid Reinforcement Learning (RL) in adversarial MDPs using simultaneous on-policy feedback from the actually executed policy and off-policy feedback from a fixed behavior policy (known in advance). The paper proposed performance guarantees for different model setups (MAB, MDP with known/unknown dynamics).

**Compliance With Llm Reviewing Policy:**

Affirmed.

**Final Justification:**

The rebuttal addressed most of my concerns.

**Key Questions For Authors:**

Please clarify the contributions of the paper compared to existing works in more detail.

Under which conditions the performance guarantee bounds are obtained?

What if the fixed behavior policy is not known in advance?

Could you provide a numerical example showing the effects at play?

**Limitations:**

yes

**Strengths And Weaknesses:**

Strengths

S1 The considered problem of deciding in adversarial MDPs is interesting and relevant.

S2 The approach to include on-policy and off-policy feedback might have potential.

S3 The paper is overall well-written and analytical bounds are derived.


Weaknesses

W1 The approach heavily builds on the works of Gabbianelli et al.

W2 The contribution of the paper compared to existing works could be clarified in more detail.

W3 The paper is technical and dense making. There is no numerical evaluation verifying and supporting the theoretical results.

---

> ### Author Rebuttal · Authors · 2026-03-30
>
> We thank the reviewer's comments and suggestions. Below, we respond to each point in detail.
>
> > **Comment 1**: contributions
>
> **Re**: Under off-policy feedback, [1, 2] employed the **biased loss/reward estimator** to obtain regret guarantees with **single-policy concentrability (SPC)** in (i) MAB and (ii) tabular MDPs with **known transitions**.
>
> In this paper, we develop new algorithms that achieve regret bounds with SPC in MDPs even when the **transition function** $P$ **is unknown**. To the best of our knowledge, this provides **an affirmative answer to the open question** explicitly raised in Sec. 6 of [1] and Sec. 5 of [2]. Moreover, we emphasize that our work is not limited to resolving this open problem. Building upon that, we further initiate and study the **hybrid setting**. Below, we focus on technical details on SPC in the off-policy seeting.
>
> Note that the step from (i) to (ii) does not introduce uncertainty in the transition dynamics. Since $P$ is known, the main algorithmic change is to perform OMD over the **occupancy-measure space** rather than the **policy space**, while the biased loss estimator provides the pessimism needed for SPC in both settings.
>
> In contrast, when $P$ is **unknown**, the learner must additionally handle uncertainty in the transition dynamics. As discussed in Sec. 4.3.1, it is highly nontrivial. In particular, the standard **confidence-set** approach appears insufficient, since it is inherently **optimistic**, whereas establishing SPC requires a suitable form of **pessimism**.
>
> To this end, we incorporate the **biased transition function** [3] into our algorithm design, which injects the appropriate pessimism for **transition uncertainty**, which enables the desired SPC guarantee.
>
> In summary, extension from **known** to **unknown** transitions is not a routine generalization. It requires **fundamentally** addressing a new source of uncertainty and carefully introducing the right form of pessimism.
>
> > **Comment 2**: the conditions our bounds hold
>
> **Re**: They hold when the environment is **oblivious**, i.e., $\\{\ell_t\\}_{t=1}^T$ is fixed before the interaction begins, as stated in Lines 97 and 307.
>
> > **Comment 3**: Unknown behavior policy
>
> **Re**: If $\pi^B$ is fixed but unknown, we can replace it with a pessimistic empirical estimate, as in Sec. 3.2 of [1]. More broadly, our unknown-transition analysis already captures the effect of unknown $\pi^B$: when $P$ is unknown, $q^B$ must be estimated anyway to construct the loss estimator, regardless of $\pi^B$ being known or not. In terms of guarantees, unknown $\pi^B$ introduces an additional additive term of order $C(\pi^C)\sqrt{T}$ in both MABs and MDPs.
>
> > **Comment 4**: numerical example
>
> **Re**: We've conducted simulations in the MAB setting. We construct a non-stochastic environment following Sec. 8 of [4]. The configurations we set include time horizon $T$, number of actions $A$, loss gap $\Delta>0$ (in all rounds, the loss of the optimal action $a^*$ is lower than that of all other actions by $\Delta$).
>
> For behavior policy $\pi^B$, we create 3 different regimes for the value of $\\pi^B(a^{\\star})$ (and the remaining prob. mass is uniform): (i) $\pi^B(a^{\\star}) = 0.5$ ($C(a^{\\star}) = 2$); (ii) $\pi^B(a^{\\star}) = 1/A$ ($C(a^{\\star}) = A$); (iii) $\pi^B(a^{\\star}) = 1/T$ ($C(\pi^*) = T$)
>
> In the table below, we report the mean of cumulative regret ($R_T(\pi^*)$) together with its standard deviation, i.e., mean $\pm$ std, over 20 trials under $T = 10^5, A = 8, \Delta = 0.0125$.
>
> | $C(\pi^*)$ | On-policy | Off-policy | Hybrid |
> |---|---:|---:|---:|
> | 2 | 832.50 $\pm$ 194.99 | 541.13 $\pm$ 158.68 | **532.82 $\pm$ 160.43** |
> | $A = 8$ | 832.50 $\pm$ 194.99 | 638.69 $\pm$ 253.26 | **607.63 $\pm$ 228.31** |
> | $T = 10^5$ | **832.50 $\pm$ 194.99** | 1248.88 $\pm$ 0.14 | 903.34 $\pm$ 240.25 |
>
> These results support the intended behavior of our hybrid alg.: when coverage is good ($C(\\pi^\star)=2, A$), the hybrid method matches or slightly improves on the off-policy alg. (note that this does not contradict our theoretical bounds, because the off-policy alg. has learning rate $\eta^{IX} = 1/\sqrt{T}$ and the regret bound is $C(\pi^C)\sqrt{T}$, while hybrid alg. improves this to $\sqrt{C(\pi^C)T}$ in good-coverage regimes) and clearly outperforms the on-policy alg.; when coverage is very poor ($C(\\pi^*)=T$), the pure off-policy method deteriorates sharply, whereas the hybrid alg. remains closer to the on-policy alg.
>
> We refer the reviewer to anonymous GitHub repository (https://anonymous.4open.science/r/ICML-2026-submission-ID-6104-Simulation-Results-2EAC) for broader configurations.
>
> **Reference**
>
> [1] Online learning with off-policy feedback.
>
> [2] Online Learning with Off-Policy Feedback in Adversarial MDPs.
>
> [3] No-regret online reinforcement learning with adversarial losses and transitions.
>
> [4] Tsallis-inf: An optimal algorithm for stochastic and adversarial bandits.

---

> > ### Author Rebuttal · Reviewer_PL5R · 2026-04-04
> >
> > I appreciate the effort in addressing the review. Given the clarification, I raise my current scores.

---

> > > ### Author Response · Authors · 2026-04-04
> > >
> > > Dear Reviewer PL5R,
> > >
> > > Thank you for raising your score. We are pleased that our response addressed your comments. We will carefully incorporate your feedback into future revisions.

---

### Decision · Program_Chairs · 2026-04-30

**Decision:**

Accept (regular)

**Comment:**

This submission investigates hybrid Reinforcement Learning (RL) in adversarial Markov Decision Processes (MDPs), where the learner simultaneously utilizes on-policy feedback from its own actions and off-policy feedback from a fixed behavior policy. The authors propose a framework designed to handle both adversarial losses and unknown transitions, providing coverage-dependent guarantees that scale with the mismatch between the behavior and comparator policies while maintaining robust worst-case performance.

The reviewers appreciated the relevance of combining on-policy and off-policy data in adversarial settings and noted the technical effort involved in the regret analysis. During the rebuttal phase, the authors provided additional numerical simulations that successfully addressed all reviewer concerns regarding the depth and scale of the empirical evaluation. Furthermore, the authors clarified the technical nuances of handling unknown transitions alongside adversarial losses, convincing the reviewers of the work's theoretical significance. The consensus is that the paper now provides a robust and valuable foundation for hybrid learning in complex adversarial environments. Consequently, the paper is recommended for acceptance.

Recommendation: Accept.